# FastEdit: Low-Rank Structured Regularization for Efficient Model Editing

## Abstract

When new knowledge emerges, it is crucial to efficiently update large language models (LLMs) to reflect the latest information. However, state-of-the-art methods widely adopted in the model editing community—such as MEMIT, EMMET, and AlphaEdit—suffer from prohibitively slow editing speeds, often taking over 15 hours to sequentially edit 5,000 facts on models like LLaMA-3-8B, making real-time updates impractical, especially as model scale increases. Moreover, they require extensive pre-computation to sample pre-edit knowledge—a step that can take over 24 hours—severely limiting their deployability. In this paper, we present **FastEdit**, a framework that leverages the intrinsic low-rank structure of FFN key spaces not only for speed but also for more effective editing. FastEdit regularizes only the low-rank primary semantic subspace—where most pre-edit knowledge resides—while leaving the remaining directions in the key space unregularized and freely editable. This design channels edits into the unregularized subspace, thereby better preserving pre-trained knowledge in the primary semantic subspace, and enables fast computation via the Sherman–Morrison–Woodbury identity. On LLaMA-3-8B, FastEdit completes 5,000 sequential edits within 4 hours and consistently achieves higher editing accuracy and stability. Moreover, it requires only a small number of pre-edit samples, drastically reducing preprocessing overhead. Our work shows that low-rank structure provides a principled way to balance editability, efficiency, and knowledge preservation.

## 1 Introduction

Large language models (LLMs) have demonstrated remarkable capabilities in understanding and generating human language (Brown et al., 2020; Touvron et al., 2023). Yet, their knowledge remains largely static after training—updating even a single fact typically requires full retraining or incurs risks of corrupting unrelated knowledge (De Cao et al., 2021; Mitchell et al., 2022a; Meng et al., 2022; Zhao et al., 2023). This rigidity poses a fundamental challenge for applications in dynamic domains such as news, medicine, or education, where models must adapt quickly and precisely to new information (Leike et al., 2023; Vellal et al., 2024).

To enable fine-grained control over model knowledge, recent work has introduced *knowledge editing*: techniques that modify specific facts through localized weight updates while preserving general behavior (Meng et al., 2023; Ramesh et al., 2024; Gupta et al., 2024b; Fang et al., 2025). While conceptually appealing, these methods face two critical bottlenecks: computational inefficiency and practical infeasibility. Most approaches rely on expensive optimization procedures—such as inverting large $d \times d$ matrices ($d$: hidden dimension)—leading to $\mathcal{O}(d^3)$ time complexity per edit (Gupta et al., 2024a; Li & Chu, 2025; Ma et al., 2025). As a result, updating thousands of facts sequentially becomes impractical, especially for large-scale models. Moreover, many methods depend on extensive pre-computation using large sets of pre-edit data to estimate representation statistics. For example, collecting and processing samples for covariance estimation on LLaMA-3-8B (Meta, 2024) can take over 24 hours, severely limiting deployability (Meng et al., 2022;

2023; Ma et al., 2025). These costs stem from treating edits as dense, unstructured operations, without leveraging the underlying geometry of the model's latent space (Aghajanyan et al., 2021; Yu & Wu, 2023).

In this work, we ask: *Can we design a model editing framework that is both principled and truly efficient—enabling fast updates with low computational and pre-editing overhead, while better preserving existing knowledge?*

We propose **FastEdit**, a structure-aware editing framework that exploits the intrinsic low-rank structure of FFN key spaces (Aghajanyan et al., 2021; Yu & Wu, 2023). FastEdit leverages this structure by applying regularization only to the primary semantic subspace—where most pre-edit knowledge resides—while leaving its complementary subspace unregularized. This design channels edits into the unregularized directions, helping to preserve core pretrained knowledge during editing, as validated by our safety metric. Computationally, it enables a closed-form update via the Sherman–Morrison–Woodbury (SMW) identity (Golub & Van Loan, 2013), avoiding $O(d^3)$ matrix inversion and reducing per-edit complexity to $\mathcal{O}(dr^2)$ time and $\mathcal{O}(dr)$ space, where $r$ is substantially smaller than $d$.

Our approach yields three key advances: (1) a principled, structure-aware regularization that safeguards core knowledge while allowing flexible editing; (2) a highly efficient closed-form update that drastically reduces computational overhead; and (3) a scalable, incremental solver for long-term knowledge integration. Experiments on counterfactual and factual editing benchmarks demonstrate that FastEdit achieves superior edit accuracy and stability, especially under massive sequential editing. On LLaMA-3-8B, it completes 5,000 sequential edits within 4 hours—compared to over 15 hours for full-rank baselines—while requiring only a small number of pre-edit samples. These results highlight that exploiting the latent low-dimensional structure of neural representations provides a viable pathway toward efficient, reliable, and scalable model editing in practical settings.

## 2 RELATED WORK

Knowledge editing aims to update specific factual knowledge in pre-trained language models without full retraining. Existing methods fall into two main paradigms. *Training-based* approaches construct tailored datasets to train auxiliary components for parameter updates. MEND (Mitchell et al., 2022a) and InstructEdit (Zhang et al., 2024) employ meta-learning to train hypernetworks that predict localized parameter modifications. SERAC (Mitchell et al., 2022b) introduces a memory-augmented architecture with a scope classifier and a counterfactual model to generate corrected outputs. T-Patcher (Huang et al., 2023) and MELO (Yu et al., 2024) insert feedforward memory modules to store and retrieve new factual associations during inference. *Memory-based* methods, a category of *training-free* editing, store edits externally and retrieve them at inference time via similarity-based lookup (Dong et al., 2022; Zheng et al., 2023; Hartvigsen et al., 2023; Jiang et al., 2024). These approaches decouple knowledge updates from model parameters, enabling efficient and reversible edits, where in-context learning is usually utilized (Bi et al., 2025). Another *training-free* paradigm is *Locate-then-edit*, which identifies and directly modifies knowledge-localized components within the model (Wang et al., 2024; Park et al., 2025; Gupta et al., 2023; Li et al., 2024a; Gupta et al., 2024b; 2025; Dai et al., 2025). ROME (Meng et al., 2022) pioneers this approach by modeling MLP layers as associative memory and applying causal tracing to guide rank-one weight updates. Subsequent work generalizes and improves this framework: MEMIT (Meng et al., 2023) and EMMET (Gupta et al., 2024b) extends ROME to batched editing for improved scalability, while AlphaEdit (Fang et al., 2025) constrains edits within the null space of existing knowledge representations to better preserve model integrity.

Several benchmarks have been introduced to assess not only the local correctness of edits but also their ability to support logical reasoning and generalization (Zhong et al., 2023; Gu et al., 2024a; Cohen et al., 2024), providing a more comprehensive evaluation of knowledge editing methods. As knowledge editing techniques advance and are applied in more complex or sequential scenarios, unintended side effects have

increasingly come to light. These include knowledge conflict and distortion (Li et al., 2024b), gradual and catastrophic forgetting during large-scale editing (Gupta et al., 2024a), attenuation of edited knowledge over time (Li & Chu, 2024), overfitting (Zhang et al., 2025) and failure in lifelong editing due to knowledge superposition in parameter space (Hu et al., 2025). To mitigate such issues, regularization strategies have been proposed. RECT (Gu et al., 2024b) restricts updates to a sparse subset of parameters, while PRUNE (Ma et al., 2025) and AdaEdit (Li & Chu, 2025) apply singular value decomposition (SVD) to preserve the dominant components of parameter changes, thereby enhancing stability and reducing interference.

## 3 PRELIMINARIES

We focus on the *locate-then-edit* framework for model editing, which aims to update specific knowledge in large language models (LLMs) by identifying and modifying relevant parameters. Recent studies have shown that factual knowledge is primarily stored in the feed-forward network (FFN) modules of Transformers (Geva et al., 2021). Further analysis via causal mediation has revealed that editing the second linear layer within the FFN of earlier Transformer blocks is particularly effective for knowledge update (Meng et al., 2022). Concretely, each such linear layer, parameterized by a weight matrix $\mathbf{W} \in \mathbb{R}^{\hat{d} \times d}$, associates input representations $\mathbf{k} \in \mathbb{R}^d$ with output vectors $\mathbf{v} \in \mathbb{R}^{\hat{d}}$, forming a key-value mapping that encodes knowledge.

To update a piece of knowledge, we seek a new output vector $\mathbf{v}'$ that produces the desired behavior. While $\mathbf{v}'$ can be learned via gradient-based optimization, the challenge lies in finding a parameter perturbation $\Delta$ such that the updated layer $\mathbf{W} + \Delta$ maps $\mathbf{k}$ to $\mathbf{v}'$, i.e., $(\mathbf{W} + \Delta)\mathbf{k} = \mathbf{v}'$. Crucially, we want this update to retain the model's original knowledge. That is, the perturbation $\Delta$ should introduce minimal interference to the model's pre-existing behavior on unrelated knowledge.

To formalize this, suppose we have $b_0$ pieces of preserved knowledge, encoded as input-output pairs $(\mathbf{K}_0, \mathbf{V}_0)$, where $\mathbf{K}_0 \in \mathbb{R}^{d \times b_0}$ and $\mathbf{V}_0 \in \mathbb{R}^{\hat{d} \times b_0}$ satisfy $\mathbf{W}\mathbf{K}_0 = \mathbf{V}_0$. Additionally, let $b_1$ new knowledge edits be represented by $(\mathbf{K}_1, \mathbf{V}_1)$, with $\mathbf{K}_1 \in \mathbb{R}^{d \times b_1}$ and $\mathbf{V}_1 \in \mathbb{R}^{\hat{d} \times b_1}$. We then formulate the update as an optimization problem that balances faithful editing with knowledge preservation (Meng et al., 2023):

$$\Delta = \arg\min_{\tilde{\Delta}} \left( \left\| (\mathbf{W} + \tilde{\Delta})\mathbf{K}_1 - \mathbf{V}_1 \right\|^2 + \left\| (\mathbf{W} + \tilde{\Delta})\mathbf{K}_0 - \mathbf{V}_0 \right\|^2 \right). \tag{1}$$

where $\|\cdot\|$ is the Frobenius norm. Using the fact that $\mathbf{W}\mathbf{K}_0 = \mathbf{V}_0$, the second term simplifies to $\left\| \tilde{\Delta}\mathbf{K}_0 \right\|^2$. Applying the normal equation (Lang, 2012), the closed-form solution (when the inverse exists) is:

$$\begin{aligned} \Delta &= \arg\min_{\tilde{\Delta}} \left( \left\| (\mathbf{W} + \tilde{\Delta})\mathbf{K}_1 - \mathbf{V}_1 \right\|^2 + \left\| \tilde{\Delta}\mathbf{K}_0 \right\|^2 \right) \\ &= (\mathbf{V}_1 - \mathbf{W}\mathbf{K}_1)\mathbf{K}_1^\top \left( \mathbf{K}_0\mathbf{K}_0^\top + \mathbf{K}_1\mathbf{K}_1^\top \right)^{-1}. \end{aligned} \tag{2}$$

This solution provides a principled way to update model parameters while preserving existing knowledge, forming the foundation of many recent model editing methods. However, the inversion usually takes a lot of time with a time complexity of $O(d^3)$, particularly for large language models (large $d$).

## 4 METHOD

### 4.1 EFFICIENT REGULARIZATION VIA LATENT STRUCTURE MODELING

We build upon the *locate-then-edit* paradigm (Meng et al., 2022; 2023) and formulate knowledge editing as a regularized optimization problem: modify the model weights $\mathbf{W}$ by an update $\Delta$ to satisfy a new knowledge

constraint $(\mathbf{K}_1, \mathbf{V}_1)$, while minimizing interference with existing knowledge $\mathbf{K}_0$. The objective is:

$$\mathcal{L} = \|(\mathbf{W} + \Delta)\mathbf{K}_1 - \mathbf{V}_1\|_F^2 + \lambda \|\Delta \mathbf{K}_0\|_F^2, \tag{3}$$

where $\lambda > 0$ is a regularization coefficient that balances the trade-off between satisfying the new knowledge and minimizing interference with existing representations. A larger $\lambda$ enforces stronger invariance over $\mathbf{K}_0$, reducing side effects at the potential cost of underfitting the edit. The term $\|\Delta \mathbf{K}_0\|_F^2$ measures how the update $\Delta$ affects existing representations. However, directly using it in optimization can be computationally expensive. To simplify this, we can naively apply the submultiplicative property:

$$\|\Delta \mathbf{K}_0\|_F^2 \leq \|\mathbf{K}_0\|_2^2 \|\Delta\|_F^2, \tag{4}$$

which decouples $\Delta$ from $\mathbf{K}_0$ and leads to a scalar-weighted Frobenius norm. While computationally efficient, this upper bound is *structurally blind*: it penalizes all directions of $\Delta$ equally, regardless of their semantic impact on the model's latent space. To design a *semantically aware* and *efficient* regularizer, we instead consider the expected influence of $\Delta$ under a structured probabilistic model of the key distribution.

**Low-Rank Plus Diagonal (LR+D) Factor Model.** Specifically, we assume that the pre-editing hidden representation $\mathbf{k}$ follows a low-rank plus diagonal (LR+D) structure (Fan et al., 2013), motivated by empirical findings that Transformer representations often lie in a low-dimensional subspace (Aghajanyan et al., 2021; Yu & Wu, 2023). We further justify the low-rank nature of these representations from a mathematical perspective; see Appendix F.1 for a formal analysis. The model is given by:

$$\mathbf{k} = \boldsymbol{\mu} + \mathbf{U}\mathbf{z} + \boldsymbol{\varepsilon}, \tag{5}$$

where $\boldsymbol{\mu} \in \mathbb{R}^d$ is the mean (assumed to be $\mathbf{0}$ after centering), $\mathbf{z} \in \mathbb{R}^{r_0}$ is a low-dimensional latent variable ($r_0 \ll d$) with $\mathbb{E}[\mathbf{z}] = \mathbf{0}$ and $\mathrm{Cov}(\mathbf{z}) = \mathbf{I}$, $\mathbf{U} \in \mathbb{R}^{d \times r_0}$ captures dominant semantic directions (e.g., topics or relations), and $\boldsymbol{\varepsilon} \sim \mathcal{N}(\mathbf{0}, \mathbf{D})$ represents isotropic or anisotropic noise with diagonal covariance $\mathbf{D} = \mathrm{diag}(d_1, \ldots, d_d)$, independent of $\mathbf{z}$. This model subsumes several important special cases. When $\mathbf{U} = \mathbf{0}$, it reduces to a diagonal-covariance Gaussian: $\mathbf{k} \sim \mathcal{N}(\boldsymbol{\mu}, \mathbf{D})$. Further, if $\mathbf{D} = \sigma^2 \mathbf{I}$, the covariance becomes isotropic ($\mathbf{C} = \sigma^2 \mathbf{I}$), and the expected penalty $\mathbb{E}\left[\|\Delta \mathbf{k}\|_F^2\right]$ becomes $\sigma^2 \|\Delta\|_F^2$, recovering the scalar-scaled Frobenius norm in Equation 4.

Under this model, the expected regularization term in Equation 3 can be derived as:

$$\mathbb{E}_{\mathbf{K}_0}\left[\|\Delta \mathbf{K}_0\|_F^2\right] \propto \mathbb{E}_{\mathbf{k}}\left[\|\Delta \mathbf{k}\|_2^2\right] = \mathrm{Trace}\left(\Delta^\top \Delta \left(\mathbf{U}\mathbf{U}^\top + \mathbf{D}\right)\right),$$

See Appendix F.2 for a detailed derivation. This expectation reveals that the impact of $\Delta$ is governed not merely by its magnitude, but by its alignment with the underlying structure of the key space. Specifically, edits that align with the semantic subspace spanned by $\mathbf{U}$—i.e., directions of high data variance—have greater influence on existing representations and are thus more disruptive. In contrast, perturbations in the orthogonal complement $\mathbf{U}^\perp$—i.e., the null space of $\mathbf{U}^\top$—affect lower-variance directions and incur less interference. Consequently, the expectation implicitly encodes the geometry of the latent representation space, assigning higher penalty to changes along semantically salient directions. Replacing the empirical Frobenius norm $\|\Delta \mathbf{K}_0\|_F^2$ in Equation 3 with this expected regularizer leads to the modified objective:

$$\Delta = \arg\min_{\hat{\Delta}} \left\|(\mathbf{W} + \hat{\Delta})\mathbf{K}_1 - \mathbf{V}_1\right\|_F^2 + \lambda \cdot \mathrm{Trace}\left(\hat{\Delta}^\top \hat{\Delta}(\mathbf{U}\mathbf{U}^\top + \mathbf{D})\right). \tag{6}$$

Letting $\mathbf{R} = \mathbf{V}_1 - \mathbf{W}\mathbf{K}_1$, the closed-form solution is given by:

$$\Delta = \mathbf{R}\mathbf{K}_1^\top \mathbf{M}^{-1}, \quad \text{where } \mathbf{M} = \mathbf{K}_1\mathbf{K}_1^\top + \lambda(\mathbf{U}\mathbf{U}^\top + \mathbf{D}). \tag{7}$$

To compute $\mathbf{M}^{-1}$ efficiently, we decouple the static prior $\lambda(\mathbf{U}\mathbf{U}^\top + \mathbf{D})$ from the dynamic edit term $\mathbf{K}_1\mathbf{K}_1^\top$. Let $\mathbf{M}_0 = \lambda(\mathbf{U}\mathbf{U}^\top + \mathbf{D})$. Since $\mathbf{M}_0$ is diagonal-plus-low-rank, its inverse can be precomputed once and reused. Applying the Sherman–Morrison–Woodbury identity to $\mathbf{M} = \mathbf{M}_0 + \mathbf{K}_1\mathbf{K}_1^\top$ yields

$$\mathbf{M}^{-1} = \mathbf{M}_0^{-1} - \mathbf{M}_0^{-1}\mathbf{K}_1\left(\mathbf{I}_{b_1} + \mathbf{K}_1^\top \mathbf{M}_0^{-1}\mathbf{K}_1\right)^{-1}\mathbf{K}_1^\top \mathbf{M}_0^{-1}. \tag{8}$$

Because $\mathbf{M}_0^{-1}$ admits a structured form that avoids explicit $d \times d$ storage, all operations scale linearly in $d$ and depend only on the small edit rank $b_1$. The per-edit time complexity is therefore $\mathcal{O}(dr_0b_1 + db_1^2 + b_1^3)$, which is efficient when $b_1 \ll d$. A detailed derivation is provided in Appendix D.

**Estimation of U and D.** Given the low-rank plus diagonal (LR+D) structure in Equation 5, the population covariance of a pre-editing key vector $\mathbf{k}$ is (see Appendix F.3 for a detailed derivation):

$$\mathrm{Cov}(\mathbf{k}) = \mathbb{E}\left[(\mathbf{k} - \boldsymbol{\mu})(\mathbf{k} - \boldsymbol{\mu})^\top\right] = \mathbf{U}\mathbb{E}[\mathbf{z}\mathbf{z}^\top]\mathbf{U}^\top + \mathbb{E}[\boldsymbol{\varepsilon}\boldsymbol{\varepsilon}^\top] = \mathbf{U}\mathbf{U}^\top + \mathbf{D}, \tag{9}$$

Since the true covariance is unknown, we estimate this structure from the sampled pre-editing keys $\mathbf{K}_0 \in \mathbb{R}^{d \times b_0}$ by computing the sample covariance $\mathbf{C}_{\mathrm{data}} = \frac{1}{b_0-1}(\mathbf{K}_0 - \hat{\mu})(\mathbf{K}_0 - \hat{\mu})^\top$, where $\hat{\mu}$ denotes the empirical mean of the pre-editing keys. However, when $b_0$ is small or the sampled keys are noisy, $\mathbf{C}_{\mathrm{data}}$ may provide a poor estimate of the true latent structure, leading to unstable or semantically misaligned regularization. To improve robustness, we incorporate a structural prior derived from the MLP down-projection weights $\mathbf{W}$, whose right singular vectors span the input directions of maximal variance for the MLP output. Let $\mathbf{W}_{\mathrm{down}} = \mathbf{P}\mathbf{S}\mathbf{Q}^\top$ be its SVD, and let $\mathbf{V}_r = \mathbf{Q}_{:,1:r_0} \in \mathbb{R}^{d \times r_0}$ denote the top-$r_0$ right singular vectors. We define the prior covariance as:

$$\mathbf{C}_{\mathrm{prior}} = \mathbf{V}_r \boldsymbol{\Lambda}_v \mathbf{V}_r^\top, \tag{10}$$

where $\boldsymbol{\Lambda}_v$ is a diagonal matrix of prior weights (e.g., identity or squared singular values). To ensure numerical compatibility, we normalize $\mathbf{C}_{\mathrm{prior}}$ such that $\|\mathbf{C}_{\mathrm{prior}}\|_F = \|\mathbf{C}_{\mathrm{data}}\|_F$. The fused covariance is:

$$\mathbf{C}_{\mathrm{fused}} = (1 - \alpha) \cdot \mathbf{C}_{\mathrm{data}} + \alpha \cdot \mathbf{C}_{\mathrm{prior}}, \quad \alpha \in [0, 1], \tag{11}$$

where $\alpha = 0$ recovers the data-driven estimate, and $\alpha = 1$ uses only the prior. Given the fused covariance, we compute its eigendecomposition $\mathbf{C}_{\mathrm{fused}} = \mathbf{P}\boldsymbol{\Lambda}\mathbf{P}^\top$, and set:

$$\mathbf{U} = \mathbf{P}_{:,1:r_0}\boldsymbol{\Lambda}_{1:r_0,1:r_0}^{1/2}, \tag{12}$$

$$\mathbf{D} = \mathrm{diag}\left(\mathbf{C}_{\mathrm{fused}} - \mathbf{U}\mathbf{U}^\top\right). \tag{13}$$

Selecting the top-$r_0$ eigenvectors is justified by the Eckart–Young–Mirsky theorem (Golub & Van Loan, 2013), which states that the truncated eigendecomposition provides the best rank-$r_0$ approximation to $\mathbf{C}_{\mathrm{fused}}$ in the Frobenius norm. This ensures that $\mathbf{U}\mathbf{U}^\top$ captures the most significant shared variation in the key space, while the diagonal $\mathbf{D}$ absorbs residual noise and idiosyncratic variations.

### 4.2 Efficient Sequential Editing via Periodic Spectral Compression

Real-world knowledge editing often occurs sequentially: new facts arrive in batches, requiring updates without reprocessing all prior data. Let $\{(\mathbf{K}_t, \mathbf{V}_t)\}_{t=1}^T$ denote a sequence of edit requests. At each step $t$, we aim to satisfy $(\mathbf{W}_{t-1} + \Delta_t)\mathbf{K}_t = \mathbf{V}_t$ (with $\mathbf{W}_0$ the initial weights), while preserving both *previously edited* and *pre-editing* knowledge. This requires computing the inverse of the following matrix:

$$\mathbf{M}_t = \sum_{i=1}^t \mathbf{K}_i\mathbf{K}_i^\top + \lambda(\mathbf{U}\mathbf{U}^\top + \mathbf{D}),$$

which generalizes the single-step matrix $\mathbf{M}$ in Equation 7. A straightforward approach would reapply the Sherman–Morrison–Woodbury (SMW) identity using all accumulated keys $\mathbf{K}_{1:t} = [\mathbf{K}_1, \ldots, \mathbf{K}_t]$ ($\mathbf{K}_{1:t}\mathbf{K}_{1:t}^\top = \sum_{i=1}^t \mathbf{K}_i\mathbf{K}_i^\top$). This results in a per-step computational cost of $O(dr_0 t)$, which grows linearly as the number of edits increases and as a result makes the update progressively more expensive. When $t$ becomes large—particularly as it approaches the dimension $d$ of the input key space—the inversion of SMW inner system scales as $O(t^3)$, causing the overall cost to approach $O(d^3)$ and effectively negating the efficiency gains from the low-rank assumption.

To maintain efficiency, we apply periodic low-rank compression: every $\tau$ incoming key vectors, we perform SVD on the accumulated keys and retain only the top singular components that preserve most of the directional energy. Let $\mathbf{K}_{\text{comp}}$ denote the compressed key matrix from previous cycles (or empty initially), and let $\mathbf{K}_{\text{all}} = [\mathbf{K}_{\text{comp}}, \mathbf{K}_{\text{buff}}]$ be the full set of keys to compress, where $\mathbf{K}_{\text{buff}}$ is the current buffer of unprocessed keys. We compute the SVD $\mathbf{K}_{\text{all}} = \mathbf{U}\mathbf{S}\mathbf{V}^{\top}$ and retain the largest $r$ components such that

$$r = \min\left\{ k : \frac{\sum_{i=1}^{k}\sigma_i^2}{\sum_{i=1}^{r'}\sigma_i^2} \geq \gamma, \ k \leq r_{\max} \right\},$$

where $\sigma_i$ are the singular values, $r' = \text{rank}(\mathbf{K}_{\text{all}})$, $\gamma \in (0, 1]$ is the energy retention threshold, and $r_{\max}$ caps the maximum rank to prevent unbounded growth (e.g., 3000). The compressed key matrix is then updated as $\mathbf{K}_{\text{comp}} \leftarrow \mathbf{U}_{:,1:r}\mathbf{S}_{1:r,1:r}$, and the buffer is reset. The accumulated keys $\mathbf{K}_{\text{all}}$ is used in the SMW updates at each step, ensuring per-step computational cost remains bounded at $O(dr_0 r_{\max})$. The full procedure is summarized in Algorithm 1.

## 5 EXPERIMENTS

### 5.1 EXPERIMENTAL SETUP

We detail the language models, baseline editing methods, benchmark datasets, and evaluation metrics used in our study. Full implementation details—including descriptions of baselines and datasets, hyperparameters, and evaluation protocols—are provided in Appendix B. Our code is available at https://anonymous.4open.science/r/FastEdit-SME.

**LLMs and Baselines.** We conduct experiments on three decoder-only language models: GPT2-XL (1.5B parameters), GPT-J (6B parameters), and LLaMA3 (8B parameters), with key vector dimensionalities of 6,400, 16,384, and 14,336, respectively. These models differ in both architecture and training data, enabling a comprehensive assessment of the generalization capability of editing methods across diverse large language models. We compare against a range of state-of-the-art locate-then-edit baselines: MEMIT (Meng et al., 2023), PMET (Li et al., 2024a), EMMET (Gupta et al., 2024b), AlphaEdit (Fang et al., 2025), RECT (Gu et al., 2024b), PRUNE (Ma et al., 2025), and AdaEdit (Li & Chu, 2025). We exclude ROME (Meng et al., 2022) as it corresponds to a special case of EMMET with batch size 1. Notably, most of these baselines perform poorly under sequential editing settings (Thede et al., 2025). Following Fang et al. (2025), we adapt all baselines to preserve previously edited knowledge, which substantially improves their performance (see Appendix A). We integrate our low-rank regularization framework into three representative baselines that employ distinct weight update rules: MEMIT, EMMET, and AlphaEdit. The resulting variants—denoted **MEMIT-F**, **EMMET-F**, and **AlphaEdit-F**—are obtained by incorporating a low-rank constraint into their respective parameter update formulations. Specifically, MEMIT-F implements the update rule in Equation 7, while the adaptations for EMMET-F and AlphaEdit-F are detailed in Appendix A.

**Datasets and Evaluation Metrics.** We evaluate our method on two standard factual editing benchmarks: ZsRE (Levy et al., 2017) and CounterFact (Meng et al., 2022). We adopt three core evaluation metrics: **Efficacy** (**Eff.\***), which measures whether the edited fact is predicted correctly; **Generality** (**Gen.\***), which assesses robustness to input paraphrases; and **Specificity** (**Spe.\***), which evaluates the preservation of unrelated knowledge. On CounterFact, we additionally report probability-based variants of these metrics—denoted **Eff.**, **Gen.**, and **Spe.**—following prior work (Meng et al., 2023; Fang et al., 2025). To further evaluate the locality, we assess models' general capabilities on the Stanford Sentiment Treebank (SST) (Socher et al., 2013), a binary sentiment classification task from the GLUE benchmark. Performance on this task is reported as accuracy, denoted by **SST**. For editing efficiency, the reported editing time spans from the first to the last edit, excluding data and model loading as well as post-editing performance evaluation.

## 5.2 EDITING EFFICACY WITH LOW-RANK REGULARIZATION

To isolate the impact of our low-rank regularization—which applies regularization only to the primary semantic subspace $\mathbf{U}$ while leaving the remaining directions freely editable—we adopt a controlled experimental setup that disables all fast-editing optimizations. Specifically, we use a large pre-edit key sample ($4 \times 10^7$ keys per layer, matching prior work (Meng et al., 2022; Fang et al., 2025)), set the prior fusion coefficient to $\alpha = 0$, and impose no rank constraint on accumulated edits. Under this configuration, the only difference between each baseline and its variant is whether low-rank regularization is applied to the pre-edit knowledge. The hyperparameter setting for the rank $r_0$ of $\mathbf{U} \in \mathbb{R}^{d \times r_0}$ is provided in Appendix B.

Table 1: Editing performance of full-rank versus low-rank regularized variants on CounterFact and ZsRE (2,000 sequential edits). The symbol ↑ indicates that higher values are better (edit efficacy), while ↕ denotes that metrics closer to the pre-edit values are preferable (edit locality). **Boldface** numbers highlight superior performance relative to their full-rank or low-rank counterparts.

| Method | | CounterFact | | | | | | ZsRE | | | |
|---|---|---|---|---|---|---|---|---|---|---|---|
| | | SST↕ | Eff.↑ | Gen.↑ | Spe.↑ | Eff*.↑ | Gen*.↑ | Spe*.↕ | SST↕ | Eff*.↑ | Gen*.↑ | Spe*.↕ |
| Pre-edited | | 77.0 | 15.4 | 18.0 | 83.3 | 0.40 | 0.60 | 13.7 | 77.0 | 27.7 | 27.1 | 27.4 |
| PMET | | 81.5 | 99.7 | 95.0 | 73.3 | 98.0 | 69.9 | 10.6 | 72.5 | 99.7 | 97.8 | 28.4 |
| RECT | | 80.5 | 99.2 | 89.0 | 77.5 | 95.2 | 49.1 | 9.30 | 74.0 | 98.7 | 92.9 | 27.6 |
| PRUNE | | 82.0 | 99.4 | 96.1 | 73.2 | 98.4 | 74.4 | 12.5 | 73.5 | 99.3 | 95.9 | 30.9 |
| AdaEdit | GPT-J | 81.5 | 99.7 | 96.0 | 72.7 | 98.5 | 71.5 | 10.6 | 73.5 | 99.5 | 94.6 | 26.4 |
| MEMIT | | **80.5** | 99.2 | 89.9 | **77.5** | 96.4 | 51.3 | 9.90 | 72.0 | 98.8 | 93.3 | **27.5** |
| MEMIT-F | | 81.5 | **99.8** | **93.4** | 76.9 | **98.8** | **61.4** | **11.8** | **73.0** | **99.8** | **96.2** | 28.2 |
| EMMET | | 83.0 | 99.6 | 94.4 | 75.3 | 98.1 | 60.6 | 10.4 | 75.0 | 99.8 | 97.2 | **28.6** |
| EMMET-F | | **81.5** | **99.7** | **95.6** | **75.8** | **98.4** | **65.8** | 10.4 | **76.5** | 99.8 | **97.7** | 28.9 |
| AlphaEdit | | **78.0** | 99.8 | 93.8 | 76.6 | 99.0 | 61.2 | 9.90 | 72.5 | 99.8 | 97.8 | **28.7** |
| AlphaEdit-F | | 81.5 | **99.8** | **94.6** | **77.8** | **99.4** | **65.4** | **10.8** | **76.0** | 99.8 | **97.9** | 28.9 |
| Pre-edited | | 96.5 | 7.80 | 10.4 | 89.3 | 0.30 | 0.50 | 21.3 | 96.5 | 38.2 | 37.6 | 38.6 |
| PMET | | 93.0 | 99.2 | 95.7 | 66.2 | 97.0 | 75.6 | 16.5 | 97.0 | 99.1 | 96.5 | 45.4 |
| RECT | | 95.0 | 99.0 | 93.0 | 71.0 | 96.8 | 71.6 | 18.5 | 96.0 | 99.1 | 96.0 | 44.0 |
| PRUNE | | 77.0 | 85.2 | 77.0 | 64.6 | 39.9 | 33.3 | 7.40 | 94.0 | 94.3 | 91.3 | 46.8 |
| AdaEdit | LLaMA3 | 63.5 | 77.6 | 70.3 | 56.1 | 36.8 | 27.7 | 0.07 | 92.5 | 93.2 | 90.6 | 46.6 |
| MEMIT | | 94.5 | 99.0 | 92.9 | 71.5 | 97.0 | **71.3** | 18.4 | 97.0 | 99.2 | 95.4 | 43.9 |
| MEMIT-F | | **96.0** | **99.6** | **93.0** | **78.1** | **98.4** | 68.4 | **20.6** | 97.0 | **99.5** | **96.0** | **43.4** |
| EMMET | | 49.0 | 51.1 | 50.8 | 48.9 | 0.00 | 0.00 | 0.00 | 93.0 | 97.3 | 93.9 | 46.0 |
| EMMET-F | | **93.0** | **98.7** | **93.9** | **68.6** | **94.2** | **69.3** | **15.6** | **94.5** | **99.4** | **96.2** | **44.6** |
| AlphaEdit | | 42.5 | 56.1 | 53.9 | 48.5 | 2.90 | 1.80 | 0.60 | 95.5 | 98.0 | 94.2 | 45.7 |
| AlphaEdit-F | | **94.0** | **98.0** | **91.5** | **66.5** | **92.8** | **69.2** | **16.8** | **96.0** | **98.8** | **95.0** | **43.5** |

Table 1 and Table 2 report editing performance under 2,000 and 5,000 sequential edits on COUNTERFACT and ZSRE, respectively. Results on GPT-2-XL and comparisons with additional baselines are included in Appendix C. Across all settings, the {○}-F variants consistently outperform their full-rank counterparts, with performance gains becoming more pronounced under 5,000 edits. Notably, both EMMET and ALPHAEDIT fail to complete 2,000 sequential edits on COUNTERFACT using LLaMA3—a model collapse phenomenon observed in prior work (Gu et al., 2024b; Gupta et al., 2024a; Fang et al., 2025; Thede et al., 2025)—whereas their low-rank regularized versions succeed. We attribute this improved stability to better preservation of the primary semantic subspace $\mathbf{U}$, quantified by the *subspace interference* metric $s_t = \|\Delta_t \mathbf{U}\|_F$, which measures the extent to which the weight update $\Delta_t$ perturbs the dominant semantic directions. Lower $s_t$

Table 2: Editing performance of full-rank vs. low-rank regularized variants under 5,000 sequential edits.

| Method | | CounterFact | | | | | | | ZsRE | | | |
|---|---|---|---|---|---|---|---|---|---|---|---|---|
| | | SST↕ | Eff.↑ | Gen.↑ | Spe.↑ | Eff*.↑ | Gen*.↑ | Spe*.↕ | SST↕ | Eff*.↑ | Gen*.↑ | Spe*.↕ |
| Pre-edited | | 77.0 | 14.7 | 17.2 | 83.5 | 0.40 | 0.50 | 14.3 | 77.0 | 27.0 | 26.2 | 27.0 |
| MEMIT | | 55.0 | 95.8 | 86.0 | 66.1 | 71.5 | 37.7 | 6.60 | 67.0 | 86.8 | 80.4 | 23.1 |
| MEMIT-F | | **74.0** | **99.2** | **89.9** | **73.5** | **94.9** | **52.6** | **9.00** | **76.0** | **96.6** | **90.5** | **26.6** |
| EMMET | GPT-J | 49.5 | 52.0 | 51.3 | 51.2 | 0.50 | 0.30 | 0.50 | **79.0** | 97.3 | 93.1 | 26.0 |
| EMMET-F | | **79.5** | **99.4** | **92.6** | 71.5 | **95.5** | **55.4** | **7.60** | 74.5 | **98.9** | **96.2** | **27.9** |
| AlphaEdit | | 79.0 | 99.3 | 90.2 | 71.9 | 95.7 | 51.9 | 7.10 | **76.0** | 99.1 | 95.0 | **27.0** |
| AlphaEdit-F | | 81.0 | **99.6** | **91.8** | **72.4** | **98.1** | **57.9** | **8.20** | 79.0 | **99.6** | **96.6** | 27.3 |
| Pre-edited | | 96.5 | 7.00 | 9.60 | 89.6 | 0.30 | 0.30 | 21.6 | 96.5 | 37.2 | 36.6 | 38.5 |
| MEMIT | | 54.0 | 95.8 | 86.0 | 66.1 | 71.5 | 37.7 | 6.60 | 95.5 | 86.8 | 80.4 | 23.1 |
| MEMIT-F | | **94.0** | **99.3** | **92.3** | **66.5** | **96.4** | **67.8** | **15.0** | **96.0** | **99.1** | **95.2** | **44.0** |
| EMMET | LLaMA3 | 50.0 | 51.1 | 50.7 | **50.0** | 0.00 | 0.00 | 0.00 | 47.5 | 1.10 | 1.10 | 0.80 |
| EMMET-F | | **50.0** | **52.2** | **51.8** | 48.9 | **0.40** | **0.40** | **0.10** | **60.5** | **75.3** | **68.7** | **23.6** |
| AlphaEdit | | 50.5 | 69.0 | 62.6 | 47.5 | 4.70 | 1.20 | 0.20 | 90.5 | 84.2 | 79.0 | **39.3** |
| AlphaEdit-F | | 50.0 | **73.4** | **65.2** | **55.0** | **34.2** | **22.6** | **6.80** | 95.5 | **97.8** | **93.7** | 44.2 |

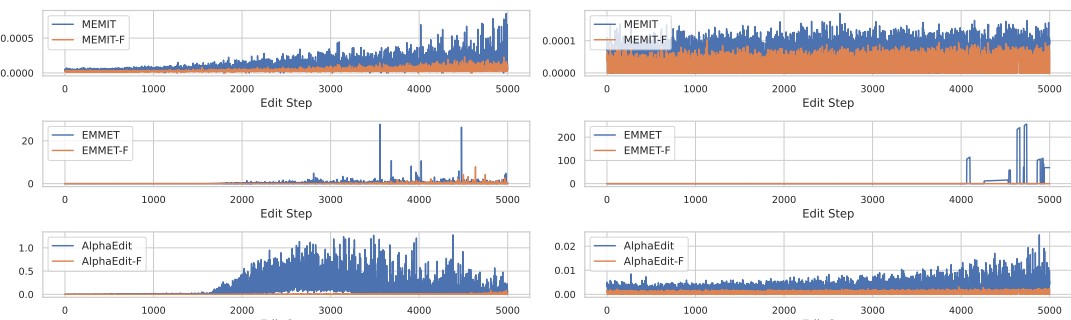

Figure 1: Temporal evolution of the edit safety metric $s_t = \|\Delta_t \mathbf{U}\|_F$ over 5,000 sequential edits on COUNTERFACT (left) and ZSRE (right) using LLaMA3. Lower $s_t$ indicates better preservation of the primary semantic subspace $\mathbf{U}$ during editing.

indicates reduced risk of corrupting existing knowledge. We evaluate $s_t$ over 5,000 sequential edits on COUNTERFACT and ZSRE using LLaMA3. As shown in Figure 1, our low-rank regularized method consistently achieves lower $s_t$ values than all baselines, demonstrating superior protection of $\mathbf{U}$ during editing. Moreover, the slower growth of $s_t$ over time reflects greater stability in the model's core knowledge representation, serving as a strong indicator of long-term editing safety. A comparison of editing mechanisms across different methods, with respect to pre-edit knowledge preservation, is provided in Appendix E.

Going further, we study the scalability of our low-rank regularization under massive sequential editing—up to 10,000 edits—on the COUNTERFACT dataset. Figure 2 shows the scaling behavior of editing methods in terms of the average of six evaluation metrics (Eff., Gen., Spe., Eff*., Gen*., and Spe*.) as the number of edits increases from 2,000 to 10,000. The results confirm that regularized variants consistently outperform their full-rank counterparts across the entire range. However, as the edit count approaches 10,000, all methods exhibit significant degradation—likely due to over-editing-induced model collapse (Gu et al., 2024b;

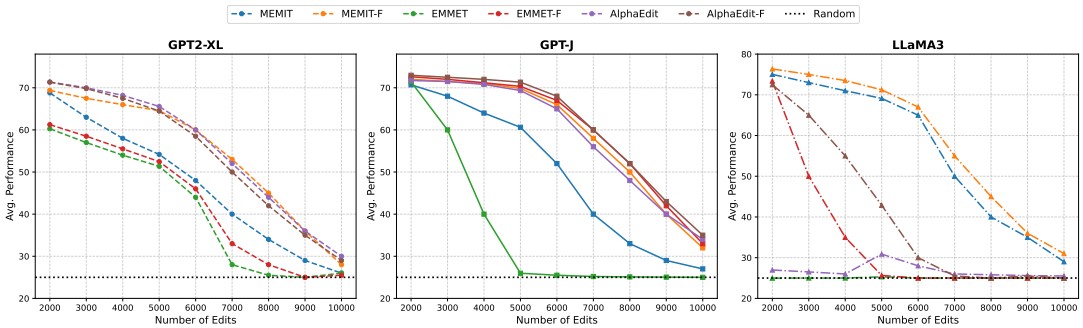

Figure 2: Editing performance on the COUNTERFACT dataset as the number of edits increases from 2k to 10k. The low-rank regularized methods consistently maintain higher performance than their full-rank counterparts.

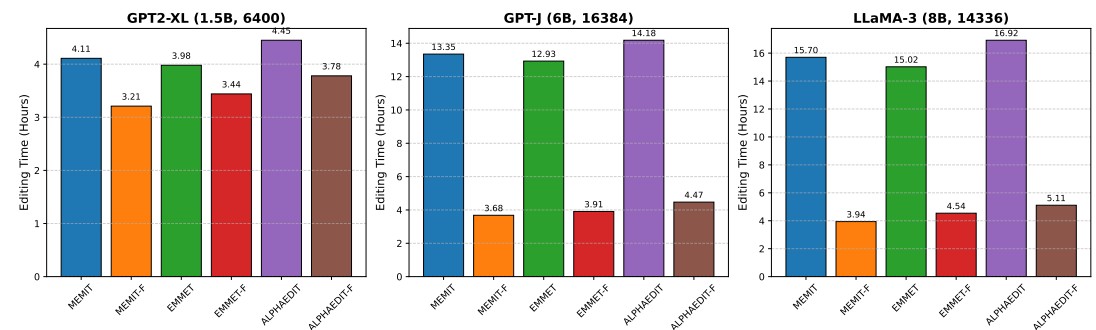

Figure 3: Editing time comparison for 5,000 sequential edits on the COUNTERFACT dataset across different models. Each subplot is labeled with the model name, parameter count, and key space dimension $d$.

Gupta et al., 2024a; Fang et al., 2025). Notably, EMMET shows the earliest performance drop, possibly because its equality constraint on edit key-value pairs introduces stronger interference into model weights; in contrast, MEMIT and ALPHAEDIT employ a squared Frobenius norm penalty, yielding smoother and more stable updates.

### 5.3 EDITING EFFICIENCY VIA LOW-RANK STRUCTURE

**Editing time.** Our method achieves substantial computational speedups by leveraging the Sherman–Morrison–Woodbury (SMW) identity to replace the expensive $O(d^3)$ matrix inversion in FFN editing with an $O(dr_0 r_{\max})$ operation. This optimization is particularly impactful for large models and multi-layer editing. Following prior work (Fang et al., 2025), we edit layers $\{13, 14, 15, 16, 17\}$ in GPT2-XL, $\{3, 4, 5, 6, 7, 8\}$ in GPT-J, and $\{4, 5, 6, 7, 8\}$ in LLaMA3. Figure 3 quantifies this speedup: on LLaMA3, FastEdit completes 5,000 sequential edits within 4 hours, compared to over 15 hours for full-rank baselines—a $4\times$ reduction in runtime. During sequential editing, the rank of the accumulated edits would naively be expected to grow linearly with the number of edits, eventually becoming the computational bottleneck. However, we show that the edited keys also exhibit a strong low-rank structure (see Appendix C) since both the pre-edit and edited keys are drawn from the same input key space. Figure 4 provides a fine-grained examination of how the maximum allowable rank $r_{\max}$ of the edited knowledge affects both editing perfor-

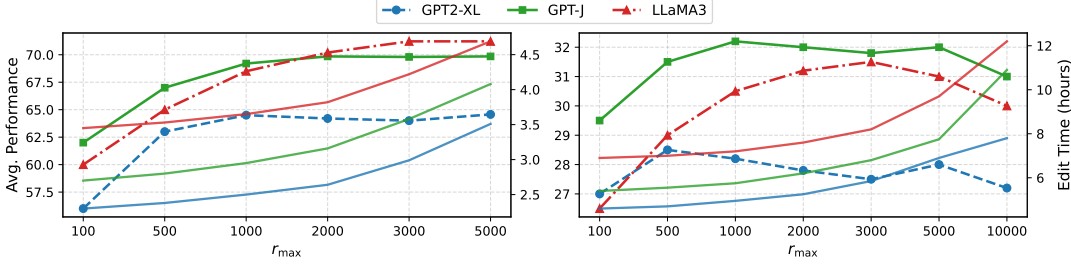

Figure 4: MEMIT-F performance (left axis, dashed lines) and editing time (right axis, solid lines) versus maximum allowable editing rank $r_{\max}$ on COUNTERFACT with 5000 edits (left) and 10000 edits (right).

mance and computational cost. We find that setting $r_{\max} \approx 2000$ is sufficient to achieve near-optimal editing accuracy for 5,000 edits. Remarkably, when scaling to 10,000 edits—where all methods exhibit significant performance degradation, as shown in Figure 2—using a *smaller* rank ($r_{\max} < 5000$) surprisingly yields better results than larger values such as $r_{\max} = 10,000$. As a result, the editing process remains highly efficient—typically completing in just a few hours for 10,000 sequential edits with small $r_{\max}$.

**Pre-computation time.** Locate-then-edit methods typically require extensive sampling of pre-editing keys (e.g., $\sim 4 \times 10^7$ in prior work (Meng et al., 2022; 2023)) to estimate the pretrained knowledge, taking over 24 hours for LLaMA3. In contrast, FastEdit requires only $\sim 4 \times 10^4$ samples, reducing pre-computation time to a few minutes. This efficiency is enabled by our robust covariance estimation procedure (detailed in Section 4), which fuses data-driven statistics with a structural prior from the MLP down-projection weights. This allows stable estimation of $\mathbf{U}$ and $\mathbf{D}$ even with limited samples, eliminating the need for massive key collection and enabling rapid deployment on new large language models. The trade-off between sample size and the fusion coefficient $\alpha$—and how the fusion improves robustness under limited sampling—is illustrated in the performance heatmaps in Appendix C.

## 6 CONCLUSION AND FUTURE WORK

We presented **FastEdit**, a model editing framework that exploits the intrinsic low-rank structure of FFN key spaces to improve both efficiency and editing quality. By applying regularization only to the primary semantic subspace and leaving the complementary directions unregularized, FastEdit channels edits into flexible dimensions while effectively preserving core pretrained knowledge. FastEdit admits a closed-form solution for weight updates and enables the use of the Sherman–Morrison–Woodbury identity to accelerate computation. On LLaMA-3-8B, FastEdit completes 5,000 sequential edits within 4 hours—compared to over 15 hours for full-rank baselines—while maintaining superior editing performance under massive sequential editing. Furthermore, by fusing pre-edit statistics with structural priors derived from MLP down-projection weights, FastEdit requires orders of magnitude fewer pre-edit samples, enabling precomputation in minutes rather than days.

**Limitations and Future Work.** FastEdit's speed advantage stems from low-rank optimization of matrix inversion in *sequential* editing task and thus diminishes in *batch* settings, where inversion is performed per batch. Nevertheless, given that sequential editing is increasingly a focus of model editing research and highly relevant to real-world applications (Thede et al., 2025; Guo et al., 2025), we believe FastEdit provides a strong and efficient baseline for this important regime. We also observe that all baseline methods, including ours, degrade beyond 10,000 sequential edits due to over-editing-induced model collapse. Sustaining performance at such scales remains a key challenge for future work.

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

# A   ADAPTATION TO LOW-RANK REGULARIZATION AND SEQUENTIAL EDITING

## A.1   ADAPTATION TO LOW-RANK REGULARIZATION

In the main text, we described how MEMIT is adapted to our low-rank regularization framework. Here, we extend this adaptation to other editing methods. Specifically, EMMET and AlphaEdit employ update rules that differ fundamentally from MEMIT's, necessitating separate derivations under the low-rank setting. In contrast, the remaining baselines—PMET, RECT, PRUNE, and AdaEdit—share the same parameter update formulation as MEMIT; consequently, their low-rank variants follow directly from the MEMIT-based adaptation and require no additional derivation. Let $\mathbf{K}_0$ and $\mathbf{K}_1 \in \mathbb{R}^{d \times r}$ denote the key matrices corresponding to the knowledge to be preserved and the knowledge to be edited, respectively.

**EMMET.** EMMET minimizes the Frobenius norm of the key perturbation subject to an exact output constraint:

$$\mathcal{L} = \|\Delta \mathbf{K}_0\|_F^2 \quad \text{s.t.} \quad (\mathbf{W} + \Delta)\mathbf{K}_1 = \mathbf{V}_1. \tag{14}$$

By modeling the pre-editing key matrix as a low-rank plus diagonal structure (as introduced in Section 4.1), the optimal update can be derived in closed form:

$$\Delta_{\text{EMMET}} = \mathbf{R}_1 \left( \mathbf{K}_1^\top \mathbf{A}^{-1} \mathbf{K}_1 \right)^{-1} \mathbf{K}_1^\top \mathbf{A}^{-1}, \tag{15}$$

where $\mathbf{R}_1 = \mathbf{V}_1 - \mathbf{W}\mathbf{K}_1$ and $\mathbf{A} = \mathbf{U}\mathbf{U}^\top + \mathbf{D}$. The matrix inversion $\mathbf{A}^{-1}$ can be computed efficiently via the Sherman–Morrison–Woodbury (SMW) identity, and the inner matrix $\mathbf{K}_1^\top \mathbf{A}^{-1} \mathbf{K}_1$ is of size $r \times r$ (with $r \ll d$), making the overall computation scalable and enabling fast editing.

**AlphaEdit.** AlphaEdit optimizes a regularized objective that balances edit accuracy, preservation of pre-editing knowledge, and weight sparsity:

$$\mathcal{L} = \|(\mathbf{W} + \Delta \mathbf{P})\mathbf{K}_1 - \mathbf{V}_1\|_F^2 + \|\Delta \mathbf{P}\mathbf{K}_0\|_F^2 + \|\Delta\|_F^2, \tag{16}$$

Again, modeling the pre-editing key matrix as a low-rank plus diagonal structure, the optimal update can be derived in closed form:

$$\Delta_{\text{AlphaEdit}} = \mathbf{R}_1 \mathbf{K}_1^T \mathbf{P}_\mathbf{U} \left( \mathbf{I_d} + \mathbf{K}_1 \mathbf{K}_1^T \mathbf{P}_\mathbf{U} \right)^{-1}, \tag{17}$$

where $\mathbf{I_d} \in \mathbb{R}^{d \times d}$ is the identity matrix and $\mathbf{P}_\mathbf{U}$ is a symmetric idempotent projection matrix ($\mathbf{P}_\mathbf{U}^\top = \mathbf{P}_\mathbf{U}$, $\mathbf{P}_\mathbf{U}^2 = \mathbf{P}_\mathbf{U}$) that projects onto the null space of $\mathbf{U}$. This design fully protects the subspace spanned by

$\mathbf{U}$ while allowing free updates in its orthogonal complement. Leveraging the low-rank structure of the edit keys and the idempotency of $\mathbf{P_U}$, the update rule can be rewritten using the SMW identity to avoid inverting large $d \times d$ matrices:

$$\Delta_{\text{AlphaEdit}} = \mathbf{R}_1 \mathbf{K}_1^\top \mathbf{P_U} - \mathbf{R}_1 \left( \mathbf{K}_1^\top \mathbf{P_U} \mathbf{K}_1 \right) \left( \mathbf{I_r} + \mathbf{K}_1^\top \mathbf{P_U} \mathbf{K}_1 \right)^{-1} \mathbf{K}_1^\top \mathbf{P_U}, \tag{18}$$

where $\mathbf{I_r} \in \mathbb{R}^{r \times r}$ is the identity matrix. The critical inversion is now reduced to an $r \times r$ system, significantly improving computational efficiency. This reformulation follows directly from the Sherman–Morrison–Woodbury identity and the projection property of $\mathbf{P_U}$.

## A.2 ADAPTATION TO SEQUENTIAL EDITING

To enable fair comparison in sequential editing scenarios, we adapt all baseline methods to preserve previously edited knowledge. Let:

- $\mathbf{K}_0$: initial model keys (pre-editing),

- $\mathbf{K}_t$: current batch of edit keys,

- $\mathbf{K}_{1:t-1} = [\mathbf{K}_1, \ldots, \mathbf{K}_{t-1}]$: keys from all previous edits.

Below we describe the adapted update rules.

**MEMIT and MEMIT-based Methods (PMET, RECT, PRUNE, AdaEdit):** The original MEMIT update is:

$$\Delta_{\text{MEMIT}} = \mathbf{R} \mathbf{K}_t^T \left( \mathbf{K}_0 \mathbf{K}_0^T + \mathbf{K}_t \mathbf{K}_t^T \right)^{-1}. \tag{19}$$

To protect previously edited knowledge, we extend the regularization term:

$$\Delta_{\text{MEMIT}} = \mathbf{R} \mathbf{K}_t^T \left( \mathbf{K}_0 \mathbf{K}_0^T + \mathbf{K}_{1:t-1} \mathbf{K}_{1:t-1}^T + \mathbf{K}_t \mathbf{K}_t^T \right)^{-1}. \tag{20}$$

This adaptation is also applied to PMET, RECT, PRUNE, and AdaEdit, as they are built upon the MEMIT framework.

**EMMET:** The original update rule is:

$$\Delta_{\text{EMMET}} = \mathbf{R}_t \left( \mathbf{K}_t^T (\mathbf{K}_0 \mathbf{K}_0^T)^{-1} \mathbf{K}_t \right)^{-1} \mathbf{K}_t^T (\mathbf{K}_0 \mathbf{K}_0^T)^{-1}. \tag{21}$$

We adapt it by updating the inverse covariance estimate to include prior edits:

$$\Delta_{\text{EMMET}} = \mathbf{R}_t \left( \mathbf{K}_t^T (\mathbf{K}_0 \mathbf{K}_0^T + \mathbf{K}_{1:t-1} \mathbf{K}_{1:t-1}^T)^{-1} \mathbf{K}_t \right)^{-1} \mathbf{K}_t^T (\mathbf{K}_0 \mathbf{K}_0^T + \mathbf{K}_{1:t-1} \mathbf{K}_{1:t-1}^T)^{-1}. \tag{22}$$

**AlphaEdit:** AlphaEdit inherently supports sequential editing by design. Its update already includes protection for previously edited knowledge:

$$\Delta_{\text{AlphaEdit}} = \mathbf{R} \mathbf{K}_t^T \mathbf{P} \left( \mathbf{I} + \mathbf{K}_{1:t-1} \mathbf{K}_{1:t-1}^T \mathbf{P} + \mathbf{K}_t \mathbf{K}_t^T \mathbf{P} \right)^{-1}, \tag{23}$$

where $\mathbf{P}$ is a projection matrix onto the null space of preserved knowledge. Hence, no further adaptation is required.

## B EXPERIMENTAL SETUP

### B.1 BASELINE MODEL EDITING METHODS

We summarize the core ideas of the following seven representative methods:

- **MEMIT** (Meng et al., 2023): Extends single-fact editing (e.g., ROME (Meng et al., 2022)) to batched updates via a least-squares optimization, enabling efficient integration of multiple facts through direct weight modification.

- **PMET** (Li et al., 2024a): Improves editing precision by analyzing information flow in Transformer layers. It observes that Multi-Head Self-Attention (MHSA) encodes general reasoning patterns and should remain unaltered. PMET optimizes hidden states of both MHSA and FFN but only uses FFN states to update weights for more targeted edits.

- **EMMET** (Gupta et al., 2024b): Unifies ROME and MEMIT under a common preservation-memorization objective. While ROME uses equality constraints for single edits, EMMET supports batched editing with the same constraint type, achieving comparable performance with theoretical consistency.

- **AlphaEdit** (Fang et al., 2025): Addresses knowledge disruption in sequential editing by projecting perturbations into the null space of preserved knowledge. This ensures that outputs on unedited facts remain unchanged, significantly improving edit retention with minimal overhead.

- **RECT** (Gu et al., 2024b): Highlights that excessive weight changes during editing degrade general capabilities (e.g., reasoning, inference). It regularizes updates based on the relative change in weights to preserve model functionality while maintaining edit success.

- **PRUNE** (Ma et al., 2025): Identifies the condition number of the edit matrix as a key factor affecting stability in sequential editing. By constraining this value, PRUNE limits parameter perturbation and preserves general knowledge over many edits.

- **AdaEdit** (Li & Chu, 2025): Tackles performance decay in continuous editing by promoting disentangled and sparse representations of edited knowledge, enabling robust performance in large-scale editing scenarios.

### B.2 EDITING DATASETS & EVALUATION METRICS.

We evaluate on two standard factual editing benchmarks: ZsRE (Levy et al., 2017) and CounterFact (Meng et al., 2022). Each edit is defined by an input-output pair $(x, y)$. In ZsRE, $x$ is a question (e.g., *What university did Watts Humphrey attend?*) and $y$ is the target answer (e.g., *Illinois Institute of Technology*). In CounterFact, $x$ is a cloze prompt (e.g., *The mother tongue of Danielle Darrieux is*) and $y$ is the new fact (e.g., *English*), with the original fact $y_o$ (e.g., *French*) provided. We assess performance using three primary metrics: **Efficacy (Eff\*)** measures whether the model generates $y$ as the top prediction given $x$, i.e., $y = \arg\max_{y'} P(y' \mid x)$. **Generality (Gen\*)** evaluates robustness to input variation by measuring success on paraphrased inputs $x_g$, i.e., $y = \arg\max_{y'} P(y' \mid x_g)$. **Specificity (Spe\*)** measures locality by the percentage of unrelated fact pairs $(x_s, y_s)$ that remain correctly predicted after editing. For CounterFact, since the original fact $y_o$ is provided, we additionally report probability-increase-based metrics, where an edit is considered successful if $P(y \mid x) > P(y_o \mid x)$, following prior work (Meng et al., 2023; Fang et al., 2025). The corresponding metrics are denoted as **Eff.**, **Gen.**, and **Spe.**.

Following prior work (Fang et al., 2025), we also evaluate the model's general capabilities on a GLUE benchmark task to assess whether its performance degrades after editing—effectively serving as an additional test of locality. Specifically, we select the Stanford Sentiment Treebank (SST) (Socher et al., 2013) as a representative task. SST is a binary sentiment classification dataset comprising sentences from movie reviews, where

each input must be classified as either positive (label = 1) or negative (label = 0). For example, the sentence *"What better message than 'love thyself' could young women of any size receive?"* is annotated with label 1 (positive). Performance on this task is reported as accuracy, denoted by **SST**. We evaluate the model in a zero-shot prompting setting using the prompt template `Review:{input}\nSentiment:`, where the model's prediction is based on the probabilities of the tokens "positive" and "negative". The evaluation is conducted on 200 SST samples.

## B.3 IMPLEMENTATION DETAILS

All experiments are conducted on an NVIDIA A6000 GPU with 48 GB of memory, using `bfloat16` precision for all models. Our implementation is built upon the official codebase of AlphaEdit (Fang et al., 2025)[1]. Specifically, we reuse their implementations for key and value computation and only modify the parameter update rule as defined in Equation 7. When evaluating the efficiency of our proposed low-rank structured regularization, the reported editing time corresponds to the duration from the first to the last edit, excluding data and model loading as well as post-editing performance evaluation.

Regarding hyperparameters, our method introduces three main components: the prior rank $r_0$ used in the low-rank regularization and the maximum rank $r_{max}$ of the accumulated editing keys $\mathbf{K}_{[1:t]}$, and the prior fusion coefficient $\alpha$.[2] The regularization strength $\lambda$ is shared with baseline methods; thus, we adopt their standard setting. Specifically, $\lambda$ is uniformly set to 15,000 across all datasets and editing models (Meng et al., 2023; Fang et al., 2025). Two implementation settings are fixed: (i) periodic compression of $\mathbf{K}_{[1:t]}$ with energy retention ratio $\gamma = 0.99$ every $\tau = 500$ edits (Section 4.2), and (ii) using $4 \times 10^4$ pre-editing knowledge samples for fast precomputation. The values of $r_0$, $r_{max}$, and $\alpha$ are set as follows: $r_0$ is specified in Table 3, while $r_{max} = 3000$ and $\alpha = 0.05$.

Table 3: Rank $r_0$ of the primary semantic subspace $\mathbf{U}$, selected per method, dataset, and model, where $r_0 = \min\left\{k : \frac{\sum_{i=1}^{k} \sigma_i^2}{\sum_{i=1}^{r'} \sigma_i^2} \geq \gamma\right\}$ with $\sigma_i$ denoting the singular values of $\mathbf{C}_{fused}$ (Eq. 11). The corresponding energy retention ratio $\gamma$ is shown in parentheses. Our $O(dr_0 r_{max})$ edit complexity enables fast editing.

| Model | CounterFact | | | ZsRE | | |
|---|---|---|---|---|---|---|
| | MEMIT-F | EMMET-F | AlphaEdit-F | MEMIT-F | EMMET-F | AlphaEdit-F |
| GPT2-XL (6400) | 1852 (0.7) | 1852 (0.7) | 1852 (0.7) | 1852 (0.7) | 1852 (0.7) | 2820 (0.8) |
| GPT-J (16384) | 5911 (0.8) | 5911 (0.8) | 5911 (0.8) | 3326 (0.7) | 3326 (0.7) | 5911 (0.8) |
| LLaMA3 (14336) | 5958 (0.8) | 9104 (0.9) | 1472 (0.5) | 3875 (0.7) | 3875 (0.7) | 1472 (0.5) |

## B.4 HYPERPARAMETER INVESTIGATION

In the main text, we have already analyzed the impact of the maximum editing rank $r_{max}$ (see Figure 4). Therefore, this subsection focuses exclusively on the hyperparameters $r_0$ and $\alpha$, which govern the low-rank primary semantic subspace and its fusion with the prior covariance matrix $\mathbf{C}_{fused}$ under limited pre-editing knowledge samples.

---

[1] https://github.com/jianghoucheng/AlphaEdit
[2] Note that $\alpha$ is only used when pre-editing knowledge samples are limited (e.g., $4 \times 10^4$ per layer). In our main experiments, we follow prior work and use abundant sampling ($\sim 4 \times 10^7$), where $\alpha = 0$ is optimal. The $\alpha > 0$ setting is intended solely for scenarios requiring fast precomputation to enable rapid editing on a new model.

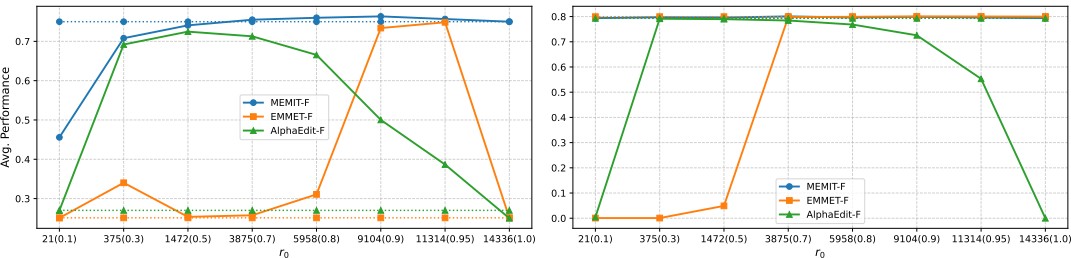

Figure 5: The impact of the rank of $\mathbf{U} \in \mathbb{R}^{d \times r_0}$, using LLaMA3 on CounterFact (left) and ZsRE (right). Each tick on the x-axis shows the selected rank $r_0$ together with its corresponding energy retention ratio $\gamma$ (in parentheses), where higher $\gamma$ indicates that more pre-editing knowledge is regularized. The performance of the full-rank regularization counterparts are shown with dashed lines.

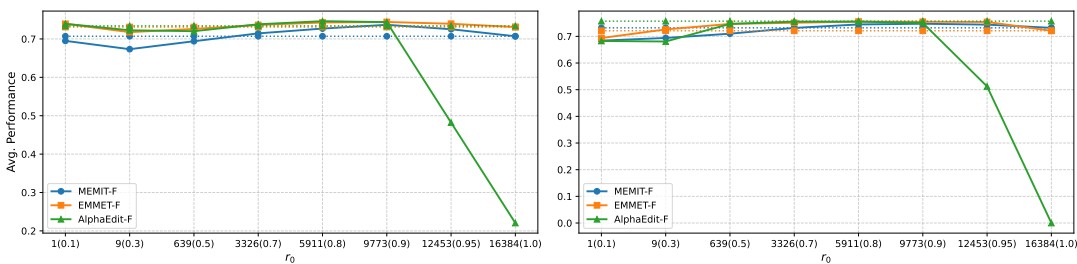

Figure 6: The impact of the rank of $\mathbf{U} \in \mathbb{R}^{d \times r_0}$, using GPT-J on CounterFact (left) and ZsRE (right).

**The impact of $r_0$.** Regarding the parameter $r_0$, the rank of the primary semantic subspace $\mathbf{U}$ is selected using the energy-based criterion introduced in Section 4.2. Specifically, we choose the smallest rank that retains at least a fraction $\gamma$ of the total energy in the fused covariance matrix $\mathbf{C}_{\text{fused}}$. Consequently, the only tunable hyperparameter in this procedure is the energy retention threshold $\gamma \in [0, 1]$. Figure 5 presents an ablation study of the three low-rank regularization variants (MEMIT-F, EMMET-F, and AlphaEdit-F) with respect to the regularization rank $r_0$—equivalently, the energy retention ratio $\gamma$—on the CounterFact and ZsRE datasets using Llama-3 under 2,000 sequential edits. The results show that MEMIT-F and EMMET-F achieve their best performance when $\gamma \in [0.7, 1.0]$, whereas AlphaEdit-F performs optimally for $\gamma \in [0.3, 0.9]$. This difference arises because AlphaEdit-F relies on null-space projection; setting $\gamma = 1$ (i.e., retaining full rank) eliminates the null space, rendering the editing mechanism ineffective. As a result, AlphaEdit-F exhibits a distinct performance landscape compared to the other methods. Notably, the original EMMET and AlphaEdit fail to sustain 2,000 sequential edits on CounterFact, while their low-rank regularization variants (EMMET-F and AlphaEdit-F) succeed when equipped with an appropriate choice of $r_0$ (or equivalently, $\gamma$). Results on GPT-J and GPT2-XL are shown in Figures 6 and 7, where similar conclusion holds.

**The impact of $\alpha$.** Pre-editing knowledge sampling is computationally demanding: prior work typically collects around $4 \times 10^7$ FFN keys per edited layer (Meng et al., 2022). Although this precomputation is performed only once per LLM, it impedes rapid adaptation to new models. To mitigate this, we reduce the sample size by up to three orders of magnitude (down to $4 \times 10^4$ keys per layer) and introduce a sensitivity-aware prior derived from the down-projection matrix of the FFN module, whose leading right singular vector identifies the most editing-sensitive direction in the input space—this direction should be preserved during updates. To balance the empirical covariance (from limited samples) and this sensitivity-based prior, we fuse

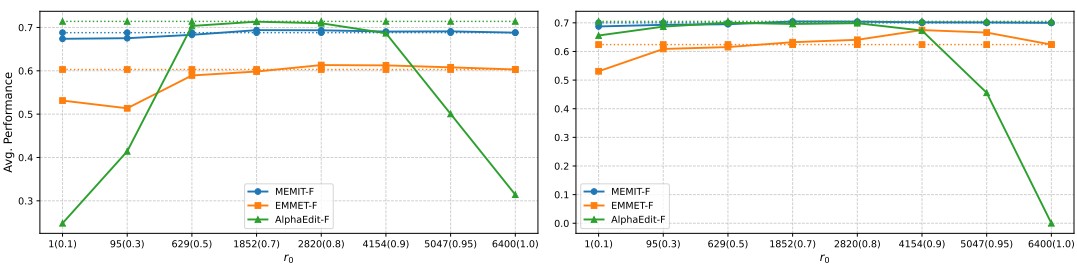

Figure 7: The impact of the rank of $\mathbf{U} \in \mathbb{R}^{d \times r_0}$, using GPT2-XL on CounterFact (left) and ZsRE (right).

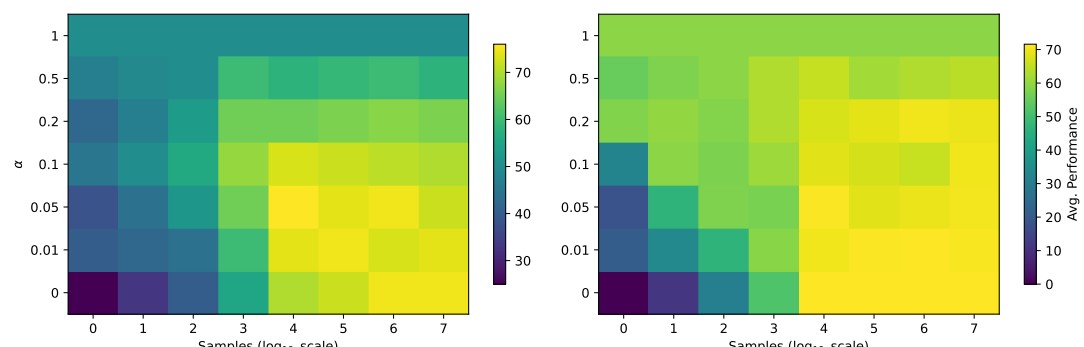

Figure 8: Heatmap of editing performance across varying sample sizes (x-axis: $0, 10, 10^2, \ldots, 10^7$) and $\alpha$ values (y-axis), using MEMIT-F under 2,000 sequential edits on LLaMA3. Results are shown for CounterFact (left) and ZsRE (right). Here, $\alpha = 0$ corresponds to using only the empirical covariance estimated from samples, while $\alpha = 1$ relies exclusively on the sensitivity direction derived from the down-projection matrix. Intermediate values ($\alpha \in [0, 0.2]$) interpolate between these two priors and yield improved editing performance when pre-edit sampling is limited.

them via the coefficient $\alpha$, where $\alpha = 0$ uses only sampled statistics and $\alpha = 1$ relies solely on the prior. Figure 8 presents a heatmap of editing performance across varying sample sizes (x-axis: $0, 10, 10^2, \ldots, 10^7$) and $\alpha$ values (y-axis). The results reveal a clear trade-off: (1) under limited sampling (e.g., $\leq 10^2$ keys), incorporating the prior consistently improves the editing performance, as the empirical covariance is too noisy to guide editing reliably; (2) under abundant sampling (e.g., $\geq 10^6$ keys), the best performance is achieved at $\alpha \approx 0$, and larger $\alpha$ values may degrade results—indicating that the data-driven estimate becomes sufficient and the fixed prior introduces bias. In view of this, we set $\alpha = 0.01$ and use $4 \times 10^4$ pre-editing samples per layer for fast precomputation and effective editing.

## C  ADDITIONAL EXPERIMENTAL RESULTS

### C.1  EDITING PERFORMANCE ON GPT2-XL

Table 4 presents the results for GPT2-XL on the CounterFact and ZsRE datasets under 2,000 and 5,000 sequential edits. The low-rank regularized versions generally outperform their full-rank counterparts, with **boldface** values indicating superior performance.

Table 4: Editing performance of full-rank versus low-rank regularized variants on GPT-2-XL under 2,000 and 5,000 sequential edits, evaluated on the CounterFact and ZsRE datasets.

| Method | | CounterFact | | | | | | ZsRE | | |
|---|---|---|---|---|---|---|---|---|---|---|
| | | Eff.↑ | Gen.↑ | Spe.↑ | Eff*.↑ | Gen*.↑ | Spe*.↕ | Eff*.↑ | Gen*.↑ | Spe*.↕ |
| Pre-edited | | 22.1 | 24.4 | 78.0 | 0.10 | 0.40 | 10.6 | 23.7 | 22.8 | 25.0 |
| MEMIT | 2000 | 98.0 | 88.6 | **65.7** | 91.4 | 58.4 | **10.6** | 94.7 | 88.4 | 26.9 |
| MEMIT-F | | **98.0** | **89.4** | 65.5 | **92.5** | **60.8** | 10.1 | **95.4** | **89.2** | **26.8** |
| EMMET | | 92.4 | 85.6 | 57.1 | 71.1 | **49.9** | 5.50 | 85.0 | 77.8 | 24.3 |
| EMMET-F | | **93.4** | **85.7** | **58.2** | **75.6** | 48.7 | **5.90** | **91.1** | **85.8** | **25.5** |
| AlphaEdit | | 99.6 | 94.0 | **65.7** | **97.3** | 65.3 | 6.40 | **96.0** | **88.8** | 26.8 |
| AlphaEdit-F | | **99.6** | **94.0** | 65.6 | 96.8 | **65.6** | **6.40** | 95.2 | 88.0 | **26.4** |
| Pre-edited | | 21.4 | 23.8 | 78.2 | 0.20 | 0.40 | 10.6 | 22.8 | 21.8 | 24.3 |
| MEMIT | 5000 | 89.6 | 76.2 | 58.0 | 60.9 | 34.1 | 6.30 | 92.3 | 84.6 | **25.3** |
| MEMIT-F | | **96.7** | **85.5** | **60.5** | **85.7** | **51.4** | **7.60** | **93.2** | **85.5** | 25.5 |
| EMMET | | 87.4 | 76.3 | 55.5 | 54.6 | 30.6 | 3.70 | 82.1 | 76.0 | **21.8** |
| EMMET-F | | **88.2** | **79.2** | **56.2** | **55.6** | **31.5** | **4.10** | **84.7** | **79.9** | 21.0 |
| AlphaEdit | | **98.0** | **88.3** | 60.8 | **88.7** | **52.9** | 4.70 | **91.2** | **84.3** | **23.6** |
| AlphaEdit-F | | 97.8 | 87.7 | **61.0** | 85.4 | 50.1 | **4.90** | 90.6 | 83.3 | 23.2 |

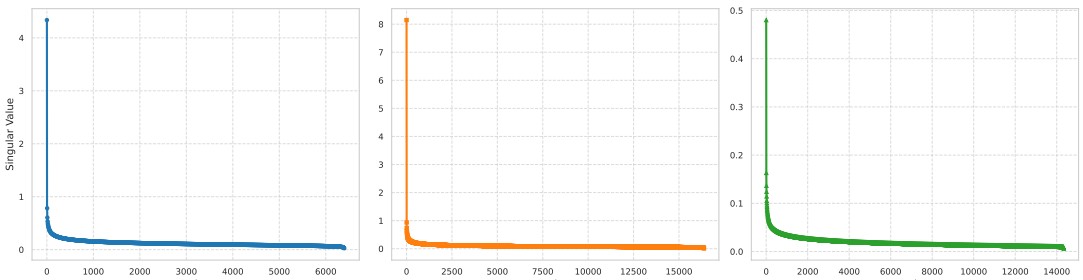

Figure 9: Singular value distribution of the accumulated editing keys $\mathbf{K}_{[1:5000]}$ for GPT-2-XL, GPT-J, and LLaMA3 (from left to right), using the CounterFact dataset and the MEMIT-F editing method.

## C.2 THE LOW-RANK STRUCTURE OF THE EDITING KEYS

We perform 5,000 sequential edits on the CounterFact dataset using MEMIT-F with GPT2-XL, GPT-J, and LLaMA3, collecting the resulting editing keys into matrices $\mathbf{K}_{[1:5000]} \in \mathbb{R}^{d \times 5000}$. We plot the singular value spectra and observe a long-tailed distribution: most of the energy is concentrated in the leading singular values, while the rest decay rapidly. As shown in Figure 9, all three key matrices exhibit a pronounced low-rank structure. This is expected: both pre-editing and post-editing keys are derived from intermediate activations of Transformer MLP layers, which are known to reside in low-dimensional subspaces (Aghajanyan et al., 2021; Yu & Wu, 2023). Consequently, even after extensive editing, the accumulated keys remain approximately low-rank—providing empirical justification for our low-rank modeling assumption and the use of a maximum allowable rank $r_{\max}$ to cap the rank of the accumulated editing keys $\mathbf{K}_{[1:t]}$.

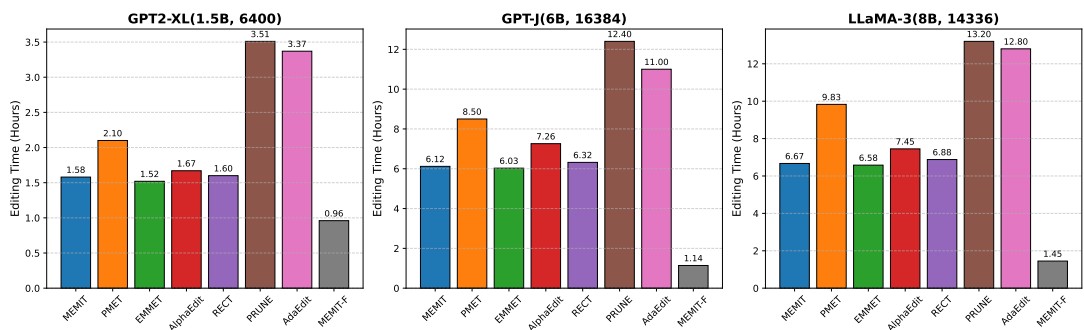

Figure 10: Editing time comparison for 2,000 sequential edits on the COUNTERFACT dataset across different models. Each subplot is labeled with the model name, parameter count, and key space dimension $d$.

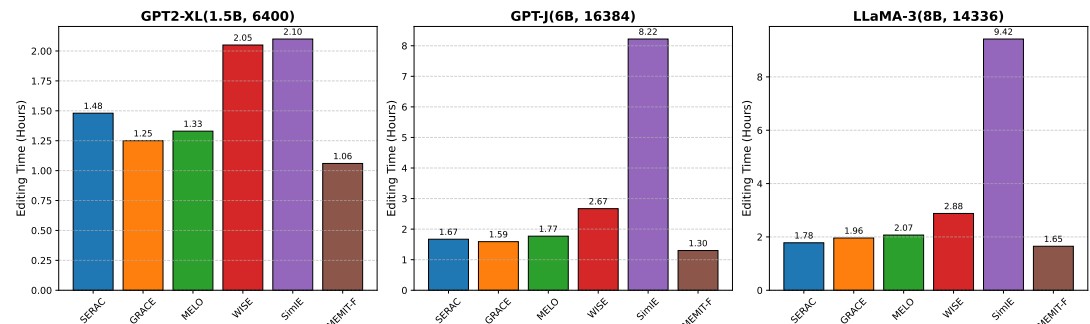

Figure 11: Editing time comparison for 2,000 sequential edits on the COUNTERFACT dataset across different models. Each subplot is labeled with the model name, parameter count, and key space dimension $d$. Timing includes all 2000 edits and the post-editing performance evaluation.

### C.3 EDITING TIME UNDER 2,000 SEQUENTIAL EDITS

Figure 10 shows the total time required to perform 2,000 sequential edits on the CounterFact dataset. Benefiting from low-rank modeling and the use of the Sherman–Morrison–Woodbury (SMW) identity, MEMIT-F achieves the fastest editing across all models—GPT2-XL (1.5B), GPT-J (6B), and LLaMA3 (8B)—with only 0.96, 1.14, and 1.45 hours, respectively. It is approximately $5\times$ faster than the next-fastest baseline on larger models and up to $10\times$ faster than PRUNE and AdaEdit.

### C.4 EDITING PERFORMANCE COMPARISON WITH MORE BASELINES

Our low-rank regularization method is grounded in a key-value modification and preservation objective (Equation 3). In this section, we compare it against baselines that do not follow this line of research. Specifically, we select SERAC (Mitchell et al., 2022b), GRACE (Hartvigsen et al., 2023), MELO (Yu et al., 2024), WISE (Wang et al., 2024), and SimIE (Guo et al., 2025) as representative baselines. SERAC, GRACE, MELO, and WISE are memory-based editing methods that employ external memory modules to store (or learn) edited facts. At inference time, they determine whether to retrieve and apply a stored (or learned) fact based on the similarity between the input query and memorized entries. SimIE is a recently proposed

method specifically designed for sequential editing. It maps the update matrix of the current edit to a new update matrix that also protects previous edits—a protection strategy distinct from ours (Appendix A).

Table 5 summarizes the editing performance of various methods under 2,000 sequential edits on the CounterFact and ZsRE benchmarks, while Figure 11 reports the end-to-end editing time (from edit request to final evaluation). Among the baselines, GRACE achieves the strongest locality—i.e., it minimally affects model predictions on unrelated inputs—yet suffers from poor generalization to paraphrased queries. This limitation stems from its memory-based design: GRACE stores key-value (KV) pairs for edited facts and, during inference, retrieves and replaces the corresponding forward activations whenever a match is detected. Although it avoids computing an explicit weight update, the retrieval and dynamic hidden activations substitution process introduces non-negligible overhead at inference time. MELO, like GRACE, is based on query-key matching, but instead of storing raw activations, it dynamically selects a learned LoRA adapter at inference time to apply the edit. In contrast, SERAC suffers from catastrophic forgetting and is therefore unsuitable for sequential editing. Both SERAC and WISE rely on trainable external modules—such as adapters or memory networks—that require additional training data for optimization. This not only increases the editing cost but also introduces computational latency during inference due to the extra forward pass through the auxiliary module. SimIE addresses sequential editing by learning a mapping from the current update matrix to a modified one that protects previous edits. While effective, this mapping operation incurs additional computation per edit.

Table 5: Performance comparison of various model editing methods across three base models (GPT2-XL, GPT-J, and LLaMA3) on the CounterFact and ZsRE datasets under 2,000 sequential edits. GRACE achieves the best locality but suffers from poor generalization. WISE, SimIE, and FastEdit offer more balanced performance, with FastEdit consistently outperforming the others while enabling the fastest editing.

| Method | | CounterFact | | | | | | ZsRE | | |
|---|---|---|---|---|---|---|---|---|---|---|
| | | Eff.↑ | Gen.↑ | Spe.↑ | Eff*.↑ | Gen*.↑ | Spe*.↕ | Eff*.↑ | Gen*.↑ | Spe*.↕ |
| Pre-edited | | 22.1 | 24.4 | 78.0 | 0.10 | 0.40 | 10.6 | 23.7 | 22.8 | 25.0 |
| SERAC | | 65.2 | 48.8 | 66.4 | 61.8 | 19.1 | 10.1 | 72.7 | 38.2 | 26.7 |
| GRACE | | **98.8** | 53.4 | **73.9** | **96.7** | 21.8 | **10.4** | 94.8 | 26.9 | **25.5** |
| MELO | GPT2-XL | 95.2 | 66.3 | 63.5 | 91.9 | 41.6 | 9.80 | 93.5 | 65.2 | 27.2 |
| WISE | | 94.0 | 72.4 | 70.0 | 89.5 | 49.4 | 10.2 | 91.0 | 72.4 | 25.9 |
| SimIE | | 96.5 | 86.0 | 68.0 | 90.1 | 58.4 | 10.0 | 92.3 | 86.6 | 27.2 |
| MEMIT-F | | 98.0 | **89.4** | 65.5 | 92.5 | **60.8** | 10.1 | **95.4** | **89.2** | 26.8 |
| Pre-edited | | 15.4 | 18.0 | 83.3 | 0.40 | 0.60 | 13.7 | 27.7 | 27.1 | 27.4 |
| SERAC | | 75.5 | 49.4 | 69.1 | 71.5 | 35.4 | 12.1 | 72.0 | 38.7 | 26.5 |
| GRACE | | 98.5 | 55.6 | **74.0** | 95.0 | 22.5 | **13.1** | 96.8 | 31.5 | **27.0** |
| MELO | GPT-J | 94.9 | 76.8 | 67.0 | 91.7 | 42.4 | 11.1 | 93.2 | 42.3 | 25.8 |
| WISE | | 96.0 | 81.2 | 72.0 | 93.6 | 61.6 | 12.5 | 97.6 | 73.4 | 28.0 |
| SimIE | | 98.9 | 94.0 | 70.1 | 97.5 | 68.0 | 11.8 | 98.2 | 95.0 | 28.5 |
| MEMIT-F | | **99.8** | **96.3** | 71.4 | **98.6** | **71.6** | 12.2 | **99.6** | **96.5** | 28.2 |
| Pre-edited | | 7.80 | 10.4 | 89.3 | 0.30 | 0.50 | 21.3 | 38.2 | 37.6 | 38.6 |
| SERAC | | 63.4 | 53.5 | 69.0 | 60.2 | 18.4 | 19.1 | 60.5 | 37.8 | 36.7 |
| GRACE | | 97.0 | 52.3 | **79.3** | 96.2 | 10.3 | **20.5** | 93.5 | 40.5 | 37.9 |
| MELO | LLaMA3 | 94.9 | 59.5 | 64.8 | 92.4 | 17.4 | 15.2 | 94.1 | 40.7 | 43.6 |
| WISE | | 95.8 | 78.4 | 76.2 | 82.3 | 47.4 | 20.1 | 95.4 | 71.2 | 39.8 |
| SimIE | | 93.4 | 87.2 | 69.9 | 90.8 | 62.4 | 17.9 | 96.3 | 92.2 | 46.3 |
| MEMIT-F | | **99.6** | **93.6** | 74.4 | **98.4** | **70.5** | 19.8 | **99.0** | **95.1** | 45.3 |

In contrast, MEMIT-F achieves the best overall performance across reliability, generalization, and locality, while requiring the least computational time. By incorporating structured low-rank regularization, MEMIT-F streamlines the weight update process of locate-then-edit approaches yet significantly outperforms them in both editing quality (Tables 1, 2, 4) and editing speed (Figures 3, 4, 10). This combination of high fidelity and efficiency makes MEMIT-F uniquely well-suited for large-scale, sequential model editing scenarios.

## D    EDIT COMPLEXITY DERIVATION

Recall that $\mathbf{M}_0 = \lambda(\mathbf{U}\mathbf{U}^\top + \mathbf{D})$, where $\mathbf{D} \in \mathbb{R}^{d \times d}$ is diagonal and $\mathbf{U} \in \mathbb{R}^{d \times r_0}$. By the SMW identity,

$$\mathbf{M}_0^{-1} = (\lambda\mathbf{D})^{-1} - (\lambda\mathbf{D})^{-1}\mathbf{U}\left(\lambda^{-1}\mathbf{I}_{r_0} + \mathbf{U}^\top(\lambda\mathbf{D})^{-1}\mathbf{U}\right)^{-1}\mathbf{U}^\top(\lambda\mathbf{D})^{-1}.$$

Although $\mathbf{M}_0^{-1}$ is dense, it can be applied to any matrix $\mathbf{X} \in \mathbb{R}^{d \times b_1}$ without forming it explicitly:

$$\mathbf{M}_0^{-1}\mathbf{X} = (\lambda\mathbf{D})^{-1}\mathbf{X} - (\lambda\mathbf{D})^{-1}\mathbf{U}\underbrace{\left(\lambda^{-1}\mathbf{I}_{r_0} + \mathbf{U}^\top(\lambda\mathbf{D})^{-1}\mathbf{U}\right)^{-1}}_{\mathcal{O}(r_0^3)}\underbrace{\left(\mathbf{U}^\top(\lambda\mathbf{D})^{-1}\mathbf{X}\right)}_{\mathcal{O}(dr_0b_1)}.$$

Where the $\mathcal{O}(r_0^3)$ operation can be precomputed and hence the dominant cost is computing $(\lambda\mathbf{D})^{-1}\mathbf{U}$ and $(\lambda\mathbf{D})^{-1}\mathbf{X}$, each $\mathcal{O}(dr_0)$ and $\mathcal{O}(db_1)$, respectively, followed by matrix multiplications costing $\mathcal{O}(dr_0b_1)$. Thus, evaluating $\mathbf{M}_0^{-1}\mathbf{K}_1$ costs $\mathcal{O}(dr_0b_1)$.

Given $\mathbf{M}_0^{-1}\mathbf{K}_1$, the remaining terms in $\mathbf{M}^{-1}$ involve:

- Forming $\mathbf{K}_1^\top\mathbf{M}_0^{-1}\mathbf{K}_1$: $\mathcal{O}(db_1^2)$,
- Inverting the $b_1 \times b_1$ matrix $\mathbf{I} + \mathbf{K}_1^\top\mathbf{M}_0^{-1}\mathbf{K}_1$: $\mathcal{O}(b_1^3)$,
- Final multiplication to apply $\mathbf{M}^{-1}$ to $\mathbf{R}\mathbf{K}_1^\top$: absorbed in $\mathcal{O}(db_1^2)$.

Summing these, the total per-edit time complexity is $\mathcal{O}(dr_0b_1 + db_1^2 + b_1^3)$, and memory usage is $\mathcal{O}(d(r_0 + b_1))$, avoiding any $\mathcal{O}(d^2)$ or $\mathcal{O}(d^3)$ operations.

## E    COMPARISON BETWEEN OUR LOW-RANK REGULARIZATION AND EXISTING EDITING METHODS

We decompose the input key space $\mathbb{R}^d$ into two orthogonal subspaces: the primary semantic subspace $\mathbf{U} \in \mathbb{R}^{d \times r_0}$, where most pre-edit keys reside, and the remaining subspace $\mathbf{V}$. Table 6 compares the editing mechanism of different editing methods in terms of pre-edit knowledge preservation. AlphaEdit is not included, as it restricts updates to the null space of $\mathbf{K}_0$, thereby avoiding any regularization loss on pre-edit knowledge. Although this strategy theoretically preserves $\mathbf{K}_0$, in practice the computed "null space" is only approximate due to numerical and optimization constraints. Consequently, after 2,000 sequential edits on COUNTERFACT with LLaMA3, AlphaEdit still causes significant degradation in general language modeling performance—as evidenced by the SST metric in Table 1. As Table 6 shows, our method explicitly regularizes only the primary semantic subspace $\mathbf{U}$ while leaving the remaining subspace $\mathbf{V}$ unregularized. This design encourages edits to be channeled into $\mathbf{V}$, thereby better preserving the core pretrained knowledge in $\mathbf{U}$. To quantify this effect, we measure the interference of the update matrix $\Delta_t$ with each subspace via the Frobenius norms:

$$s_t = \|\Delta_t\mathbf{U}\|_F, \quad \bar{s}_t = \|\Delta_t\mathbf{V}\|_F, \quad s_t^* = s_t + \bar{s}_t.$$

Figures 1, 12, and 13 illustrate these quantities over 5,000 edits. We observe that our method consistently suppresses updates in $\mathbf{U}$ (i.e., smaller $s_t$), confirming better protection of the primary semantic directions.

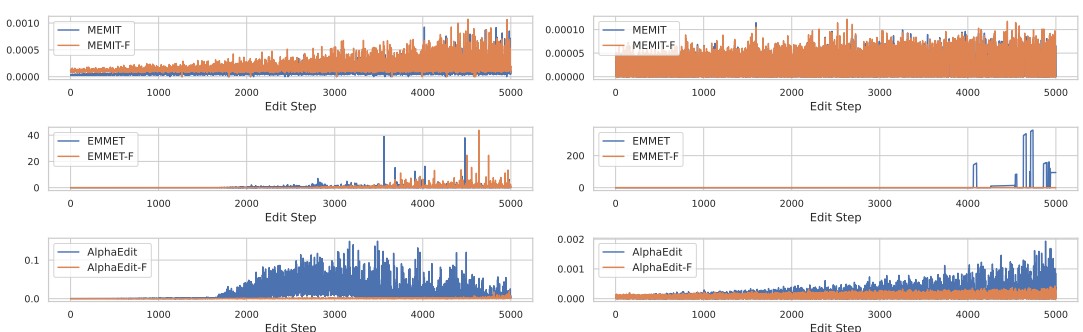

Figure 12: Temporal evolution of the edit safety metric $\bar{s}_t = \|\Delta_t \mathbf{V}\|_F$ over 5,000 sequential edits on COUNTERFACT (left) and ZSRE (right) using LLaMA3.

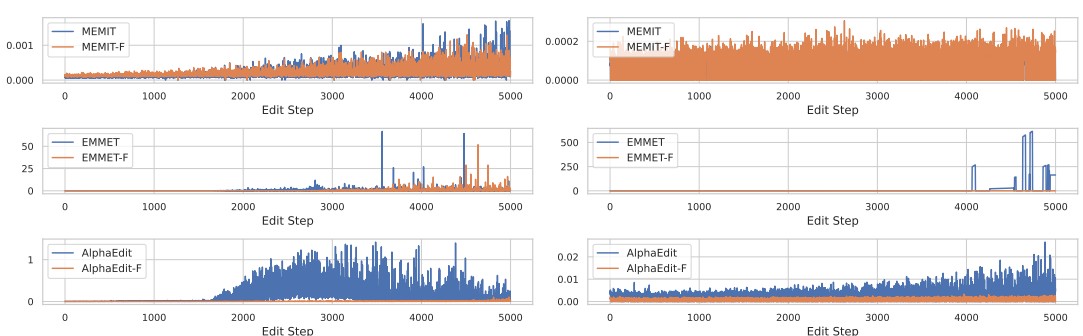

Figure 13: Temporal evolution of the edit safety metric $s_t^* = s_t + \bar{s}_t$ over 5,000 sequential edits on COUNTERFACT (left) and ZSRE (right) using LLaMA3.

Crucially, preserving $\mathbf{U}$ is more important than preserving $\mathbf{V}$: since $\mathbf{U}$ captures the bulk of pretrained knowledge, its stability delays overall model collapse, which in turn indirectly helps maintain the integrity of $\mathbf{V}$ as well.

## F  MODELING THE PRE-EDITING KEYS VIA LR+D FACTOR MODEL

### F.1  RANK ANALYSIS OF THE SECOND LINEAR LAYER INPUT

In this section, we provide a mathematical justification for the low-rank structure observed in the input to the second linear layer of the Feed-Forward Network (FFN) in Transformers. Specifically, we show that due to architectural constraints, the effective dimensionality of these activations is inherently limited.

Consider a single token's hidden representation $\mathbf{x} \in \mathbb{R}^d$ as input to the FFN block. The FFN applies the following transformation:

$$\text{FFN}(\mathbf{x}) = \mathbf{W}_2 \cdot \text{ReLU}(\mathbf{W}_1 \mathbf{x} + \mathbf{b}_1) + \mathbf{b}_2, \tag{24}$$

where $\mathbf{W}_1 \in \mathbb{R}^{4d \times d}$ and $\mathbf{W}_2 \in \mathbb{R}^{d \times 4d}$ are weight matrices, and $\mathbf{b}_1, \mathbf{b}_2$ are biases (we omit biases for simplicity in the analysis below).

Table 6: Editability and edit loss in the input key space, split into primary ($\mathbf{U}$) and remaining ($\mathbf{V}$) directions. AlphaEdit is omitted as it edits only in the (approximate) null space of $\mathbf{K}_0$, avoiding regularization in theory but not in practice; MEMIT-based baselines (e.g., RECT) are also excluded.

| Method | Editable? | | Edit Loss | |
|--------|-----------|------|-----------|------|
| | **U** | **V** | **U** | **V** |
| MEMIT | Yes | Yes | $\|\Delta\mathbf{U}\|$ | $\|\Delta\mathbf{V}\|$ |
| EMMET | Yes | Yes | $\|\Delta\mathbf{U}\|$ | $\|\Delta\mathbf{V}\|$ |
| Ours | Yes | Yes | $\|\Delta\mathbf{U}\|$ | 0 |

Let $\mathbf{a} = \text{ReLU}(\mathbf{W}_1\mathbf{x}) \in \mathbb{R}^{4d}$ denote the activation vector that serves as input to the second linear layer ($\mathbf{W}_2$). We analyze the rank properties of the set of all possible such activations.

**Linear Transformation Stage.** The first stage computes $\mathbf{z} = \mathbf{W}_1\mathbf{x}$. Since $\mathbf{W}_1 \in \mathbb{R}^{4d \times d}$, its column space (image) satisfies:

$$\dim\left(\text{Im}(\mathbf{W}_1)\right) = \text{rank}(\mathbf{W}_1) \leq d. \tag{25}$$

Thus, $\mathbf{z} = \mathbf{W}_1\mathbf{x}$ lies in a subspace of $\mathbb{R}^{4d}$ with dimension at most $d$, regardless of the specific $\mathbf{x} \in \mathbb{R}^d$.

**Nonlinear Activation Stage.** The ReLU function $\text{ReLU}(\cdot) = \max(\cdot, 0)$ is applied element-wise to $\mathbf{z}$, yielding $\mathbf{a} = \text{ReLU}(\mathbf{z})$. While ReLU is nonlinear and breaks the linear subspace structure, it does not increase the intrinsic dimensionality of the mapping. Specifically:

The image of the map $f : \mathbb{R}^d \to \mathbb{R}^{4d}$, defined by $f(\mathbf{x}) = \text{ReLU}(\mathbf{W}_1\mathbf{x})$, has topological dimension at most $d$.

*Proof.* Since $\mathbf{W}_1$ is a linear map from $\mathbb{R}^d$ to $\mathbb{R}^{4d}$, it is continuous and its image is contained in a $d$-dimensional subspace. The ReLU function is continuous and piecewise linear. The composition $f = \text{ReLU} \circ \mathbf{W}_1$ is therefore a continuous map from a $d$-dimensional domain to $\mathbb{R}^{4d}$. By standard results in topology, the image of a $d$-dimensional manifold under a continuous map cannot exceed $d$ in topological dimension. Hence, the set $\{f(\mathbf{x}) \mid \mathbf{x} \in \mathbb{R}^d\}$ forms a $d$-dimensional (or lower) subset of $\mathbb{R}^{4d}$, i.e., a $d$-dimensional manifold. $\square$

**Implication for Batched Inputs.** Now consider a batch of $n$ inputs $\{\mathbf{x}_1, \ldots, \mathbf{x}_n\}$, and let $\mathbf{A} = [\mathbf{a}_1, \ldots, \mathbf{a}_n] \in \mathbb{R}^{4d \times n}$ be the matrix of activations, where $\mathbf{a}_i = \text{ReLU}(\mathbf{W}_1\mathbf{x}_i)$. Then:

$$\text{rank}(\mathbf{A}) \leq d, \tag{26}$$

since each column $\mathbf{a}_i$ is determined by $\mathbf{x}_i \in \mathbb{R}^d$, and the mapping is deterministic. Even if $n > d$, the rank cannot exceed $d$ due to the bottleneck imposed by the input dimension and the fixed $\mathbf{W}_1$.

**Conclusion.** This analysis shows that the input to the second linear layer, $\mathbf{a}$, is constrained to a low-dimensional manifold in $\mathbb{R}^{4d}$, with intrinsic dimension at most $d \ll 4d$. This structural property justifies our use of a low-rank plus diagonal (LR+D) model for the covariance of these activations in Section 4, as the full $4d$-dimensional space is not fully utilized.

## F.2 EXPECTED REGULARIZATION TERM: DETAILED DERIVATION

In this section, we derive the expected value of the empirical regularization term $\|\Delta \mathbf{K}_0\|_F^2$ under the assumption that the pre-edit keys $\mathbf{K}_0 \in \mathbb{R}^{d \times b_0}$ are independently sampled from a zero-mean distribution with covariance structure $\mathbf{\Sigma} = \mathbf{U}\mathbf{U}^\top + \mathbf{D}$, as defined in the LR+D model (Equation equation 5).

Let $\mathbf{K}_0 = [\mathbf{k}_1, \ldots, \mathbf{k}_{b_0}]$, where each $\mathbf{k}_i \in \mathbb{R}^d$ is an i.i.d. sample satisfying:

$$\mathbb{E}[\mathbf{k}_i] = 0, \quad \mathbb{E}[\mathbf{k}_i \mathbf{k}_i^\top] = \mathbf{\Sigma} = \mathbf{U}\mathbf{U}^\top + \mathbf{D}.$$

We aim to compute:

$$\mathbb{E}_{\mathbf{K}_0}\left[\|\Delta \mathbf{K}_0\|_F^2\right],$$

where $\Delta \in \mathbb{R}^{d \times d}$ is a fixed update matrix.

The squared Frobenius norm can be expanded column-wise:

$$\|\Delta \mathbf{K}_0\|_F^2 = \sum_{i=1}^{b_0} \|\Delta \mathbf{k}_i\|_2^2.$$

Taking expectation over $\mathbf{K}_0$ (which is equivalent to taking expectation over each $\mathbf{k}_i$ due to independence):

$$\mathbb{E}_{\mathbf{K}_0}\left[\|\Delta \mathbf{K}_0\|_F^2\right] = \mathbb{E}_{\mathbf{K}_0}\left[\sum_{i=1}^{b_0} \|\Delta \mathbf{k}_i\|_2^2\right] = \sum_{i=1}^{b_0} \mathbb{E}_{\mathbf{k}_i}\left[\|\Delta \mathbf{k}_i\|_2^2\right].$$

Since all $\mathbf{k}_i$ are identically distributed, we denote $\mathbf{k} \sim p(\mathbf{k})$ as a generic key vector, and write:

$$\mathbb{E}_{\mathbf{K}_0}\left[\|\Delta \mathbf{K}_0\|_F^2\right] = b_0 \cdot \mathbb{E}_{\mathbf{k}}\left[\|\Delta \mathbf{k}\|_2^2\right].$$

Now, observe that:

$$\|\Delta \mathbf{k}\|_2^2 = (\Delta \mathbf{k})^\top (\Delta \mathbf{k}) = \mathbf{k}^\top \Delta^\top \Delta \mathbf{k}.$$

Thus,

$$\mathbb{E}_{\mathbf{k}}\left[\|\Delta \mathbf{k}\|_2^2\right] = \mathbb{E}_{\mathbf{k}}\left[\mathbf{k}^\top \Delta^\top \Delta \mathbf{k}\right].$$

We now apply the following standard result for the expectation of a quadratic form:

[Expectation of Quadratic Form] For a random vector $\mathbf{x} \in \mathbb{R}^d$ with mean $\boldsymbol{\mu}$ and covariance $\mathbf{\Sigma}$, and a fixed symmetric matrix $\mathbf{A} \in \mathbb{R}^{d \times d}$, we have:

$$\mathbb{E}[\mathbf{x}^\top \mathbf{A} \mathbf{x}] = \mathrm{Tr}(\mathbf{A}\mathbf{\Sigma}) + \boldsymbol{\mu}^\top \mathbf{A} \boldsymbol{\mu}.$$

In our case: - $\mathbf{x} = \mathbf{k}$, - $\mathbf{A} = \Delta^\top \Delta$ (symmetric and positive semi-definite), - $\boldsymbol{\mu} = \mathbb{E}[\mathbf{k}] = 0$, - $\mathbf{\Sigma} = \mathbf{U}\mathbf{U}^\top + \mathbf{D}$.

Therefore,

$$\mathbb{E}_{\mathbf{k}}\left[\mathbf{k}^\top \Delta^\top \Delta \mathbf{k}\right] = \mathrm{Tr}\left(\Delta^\top \Delta \cdot (\mathbf{U}\mathbf{U}^\top + \mathbf{D})\right) + 0^\top(\cdots)0 = \mathrm{Tr}\left(\Delta^\top \Delta\left(\mathbf{U}\mathbf{U}^\top + \mathbf{D}\right)\right).$$

Substituting back, we obtain:

$$\mathbb{E}_{\mathbf{K}_0}\left[\|\Delta \mathbf{K}_0\|_F^2\right] = b_0 \cdot \mathrm{Tr}\left(\Delta^\top \Delta\left(\mathbf{U}\mathbf{U}^\top + \mathbf{D}\right)\right).$$

This shows that the expected regularization term is proportional to the trace expression, with proportionality constant $b_0$ (the number of pre-edit keys). In practice, this constant is absorbed into the regularization coefficient $\lambda$ when forming the final objective, yielding the effective regularizer:

$$\mathrm{Tr}\left(\Delta^\top \Delta \left(\mathbf{U}\mathbf{U}^\top + \mathbf{D}\right)\right).$$

This derivation justifies replacing the empirical term $\|\Delta \mathbf{K}_0\|_F^2$ with the expected regularizer in the main optimization objective.

### F.3 Derivation of the Covariance Structure

Under the factor model $\mathbf{k} = \boldsymbol{\mu} + \mathbf{U}\mathbf{z} + \boldsymbol{\varepsilon}$ equation 5, with $\mathbb{E}[\mathbf{z}] = \mathbf{0}$, $\mathrm{Cov}(\mathbf{z}) = \mathbf{I}$, $\boldsymbol{\varepsilon} \sim \mathcal{N}(\mathbf{0}, \mathbf{D})$, and $\mathbf{z} \perp \boldsymbol{\varepsilon}$, the centered key vector is $\mathbf{k} - \boldsymbol{\mu} = \mathbf{U}\mathbf{z} + \boldsymbol{\varepsilon}$. The covariance is:

$$\mathrm{Cov}(\mathbf{k}) = \mathbb{E}\left[(\mathbf{U}\mathbf{z} + \boldsymbol{\varepsilon})(\mathbf{U}\mathbf{z} + \boldsymbol{\varepsilon})^\top\right] \tag{27}$$

$$= \mathbb{E}\left[\mathbf{U}\mathbf{z}\mathbf{z}^\top\mathbf{U}^\top\right] + \mathbb{E}\left[\mathbf{U}\mathbf{z}\boldsymbol{\varepsilon}^\top\right] + \mathbb{E}\left[\boldsymbol{\varepsilon}\mathbf{z}^\top\mathbf{U}^\top\right] + \mathbb{E}\left[\boldsymbol{\varepsilon}\boldsymbol{\varepsilon}^\top\right]. \tag{28}$$

Using independence and zero means:

- $\mathbb{E}[\mathbf{U}\mathbf{z}\mathbf{z}^\top\mathbf{U}^\top] = \mathbf{U}\,\mathbb{E}[\mathbf{z}\mathbf{z}^\top]\,\mathbf{U}^\top = \mathbf{U}\mathbf{U}^\top$,
- $\mathbb{E}[\mathbf{U}\mathbf{z}\boldsymbol{\varepsilon}^\top] = \mathbf{U}\,\mathbb{E}[\mathbf{z}]\mathbb{E}[\boldsymbol{\varepsilon}^\top] = \mathbf{0}$,
- $\mathbb{E}[\boldsymbol{\varepsilon}\mathbf{z}^\top\mathbf{U}^\top] = \mathbb{E}[\boldsymbol{\varepsilon}]\mathbb{E}[\mathbf{z}^\top]\mathbf{U}^\top = \mathbf{0}$,
- $\mathbb{E}[\boldsymbol{\varepsilon}\boldsymbol{\varepsilon}^\top] = \mathbf{D}$.

Summing the terms yields:

$$\mathrm{Cov}(\mathbf{k}) = \mathbf{U}\mathbf{U}^\top + \mathbf{D}, \tag{29}$$

as desired.

## G Algorithmic Details of Periodic Spectral Compression

We provide the full algorithmic description of the periodic spectral compression method used in our sequential knowledge editing framework. This procedure maintains an efficient low-rank approximation of accumulated edit keys by periodically compressing them via truncated singular value decomposition (SVD). Specifically, incoming key vectors are accumulated until their total dimension reaches a threshold $\tau$ (e.g., 500), at which point a global SVD is performed and only the top singular components—preserving at least a fraction $\gamma$ (e.g., 0.99) of the total energy—are retained, up to a maximum rank $r_{\max}$ (e.g., 3000). The compressed key matrix is then used in subsequent Sherman–Morrison–Woodbury (SMW) updates to ensure that each editing step remains computationally efficient, with cost bounded by $O(d r_0 r_{\max})$. The complete procedure is summarized in Algorithm 1.

---

**Algorithm 1** Efficient Sequential Editing with Periodic Spectral Compression

---

**Require:** Initial weight matrix $\mathbf{W}_0$; prior low-rank basis $\mathbf{U} \in \mathbb{R}^{d \times r_0}$ and diagonal $\mathbf{D} \in \mathbb{R}^{d \times d}$; regularization $\lambda > 0$; compression interval $\tau$; energy retention threshold $\gamma \in (0, 1]$; maximum rank $r_{\max}$.

**Ensure:** Updated weights $\mathbf{W}_T$ after $T$ sequential edits.

1: Precompute static prior matrix $\mathbf{M}_0 = \lambda(\mathbf{U}\mathbf{U}^\top + \mathbf{D})$ and its structured inverse $\mathbf{M}_0^{-1}$.

2: Initialize $\mathbf{K}_{\text{comp}} \leftarrow [\,]$, $\mathbf{K}_{\text{buff}} \leftarrow [\,]$, and $t \leftarrow 0$.

3: **for** $t = 1$ to $T$ **do**

4:     Receive edit batch $(\mathbf{K}_t, \mathbf{V}_t)$.

5:     Append to buffer: $\mathbf{K}_{\text{buff}} \leftarrow [\mathbf{K}_{\text{buff}}, \mathbf{K}_t]$.

6:     Form active key matrix:

$$\mathbf{K}_{\text{all}} \leftarrow \begin{cases} \mathbf{K}_{\text{buff}} & \text{if } \mathbf{K}_{\text{comp}} \text{ is empty,} \\ [\mathbf{K}_{\text{comp}}, \ \mathbf{K}_{\text{buff}}] & \text{otherwise.} \end{cases}$$

7:     Compute $\mathbf{M}^{-1}$ via SMW identity (Eq. equation 8):

$$\mathbf{M}^{-1} = \mathbf{M}_0^{-1} - \mathbf{M}_0^{-1}\mathbf{K}_{\text{all}} \left(\mathbf{I} + \mathbf{K}_{\text{all}}^\top \mathbf{M}_0^{-1} \mathbf{K}_{\text{all}}\right)^{-1} \mathbf{K}_{\text{all}}^\top \mathbf{M}_0^{-1}.$$

8:     Compute update: $\Delta_t = (\mathbf{V}_t - \mathbf{W}_{t-1}\mathbf{K}_t)\mathbf{K}_t^\top \mathbf{M}^{-1}$.

9:     Update model: $\mathbf{W}_t \leftarrow \mathbf{W}_{t-1} + \Delta_t$.

10:     **if** $t \bmod \tau = 0$ **or** $t = T$ **then**

11:         Perform spectral compression on $\mathbf{K}_{\text{all}}$:

12:         Compute thin SVD: $\mathbf{K}_{\text{all}} = \mathbf{U}_s \mathbf{S}_s \mathbf{V}_s^\top$.

13:         Determine retained rank:

$$r = \min \left\{ k : \frac{\sum_{i=1}^{k} \sigma_i^2}{\sum_{i=1}^{r'} \sigma_i^2} \geq \gamma, \ k \leq r_{\max} \right\},$$

    where $r' = \text{rank}(\mathbf{K}_{\text{all}})$ and $\sigma_i$ are singular values.

14:         Update compressed keys: $\mathbf{K}_{\text{comp}} \leftarrow \mathbf{U}_{s,:,1:r}\mathbf{S}_{s,1:r,1:r}$.

15:         Reset buffer: $\mathbf{K}_{\text{buff}} \leftarrow [\,]$.

16:     **end if**

17: **end for**

18: **return** $\mathbf{W}_T$

---

