# OpenReview forum: "FastEdit: Low-Rank Structured Regularization for Efficient Model Editing"
_ICLR.cc/2026/Conference — ICLR 2026 Conference Withdrawn Submission_

### Official Review · Reviewer_mcCT · 2025-10-17

**Soundness:** 4
**Presentation:** 2
**Contribution:** 2
**Rating:** 2
**Confidence:** 4

**Summary:**

This paper proposes **FastEdit**, a model-editing framework that introduces a **low-rank plus diagonal (LR+D) structured regularization** for efficient and stable knowledge updates in large language models (LLMs).
By leveraging the **Sherman–Morrison–Woodbury (SMW) identity**, the method reduces the cubic-time matrix inversion in existing editing frameworks (e.g., MEMIT, PRUNE, AlphaEdit) to a low-rank \(O(dr^2)\) computation.
A periodic spectral compression strategy is further introduced for sequential edits to maintain bounded rank and computational cost.
Experiments on GPT2-XL, GPT-J, and LLaMA-3 show up to **10× faster editing** with comparable factual accuracy.

**Strengths:**

The paper addresses an important bottleneck — the inefficiency of model editing — and offers a structured, implementable solution. FastEdit achieves order-of-magnitude acceleration (5×–10×) and memory reduction (17GB vs. 22GB) across three large models without harming edit precision.The use of the LR+D covariance model and the Sherman–Morrison–Woodbury identity is mathematically sound and clearly derived. The closed-form update (Eq.7) is clean and easy to reproduce. The writing is clear, and the method is straightforward to implement, making it useful for practitioners seeking faster model editing.

**Weaknesses:**

1. **The claimed acceleration may be overstated.**

   The reported “10× speedup” considers only the matrix inversion phase but not the **entire editing pipeline**, including the computation of  \(v\) and the LR+D covariance estimation step. When these additional costs are included, the overall acceleration is likely to be much smaller.


2. **The observed improvement mainly stems from SMW algebraic simplification.**
   The acceleration is primarily achieved through the **Sherman–Morrison–Woodbury (SMW)** identity, a standard algebraic transformation widely used in low-rank approximation and online inverse computations.
   Hence, the performance gain should be interpreted as an **engineering optimization**, rather than a genuine algorithmic innovation.


3. **The safety metrics (eₜ, sₜ) are not novel.**
   Similar orthogonality-based interference measures were already discussed in previous works.
   The metrics in FastEdit share nearly identical definitions, differing mostly in naming and visualization.

4. **Compression advantage is marginal in batch editing scenarios.**
   The claimed benefit of periodic spectral compression mainly appears in **sequential single editing**, where many edits are applied one-by-one.
   However, in **batch editing** (e.g., editing 100 facts simultaneously as in AlphaEdit), both MEMIT and FastEdit may exhibit similar runtime reductions, suggesting that compression contributes little to efficiency in such cases.

**Questions:**

See Above

---

> ### Author Response · Authors · 2025-11-27
>
> We sincerely thank the reviewer for their thoughtful and constructive feedback. Below, we address each concern in detail.
>
> **Weakness 1**
>
> The reported editing time spans from the first to the last edit, excluding data/model loading and post-editing evaluation (as clarified in Appendix B.3). The per-edit cost comprises three steps: (1) computing the key $k$, (2) computing the value $v$, and (3) computing the update matrix. Step (1) is negligible in cost. Step (2) is performed **only** at the final editing layer, whereas step (3) must be executed for **every** edited layer (e.g., 6 layers for GPT-J, following standard practice). Consequently, the dominant bottleneck lies in the matrix inversion required in step (3), which scales as $O(d^3)$ with respect to the hidden dimension $d$. While this remains tractable for GPT2-XL ($d = 6400$), it becomes prohibitively expensive for larger models such as GPT-J ($d = 16384$) and LLaMA3 ($d = 14336$). The reported speed may appear surprisingly fast—e.g., completing 5,000 sequential edits on LLaMA3 in under 4 hours—but please note that we use the **bfloat16** precision. For 6–8B-scale models like LLaMA3, bfloat16 substantially boosts computational throughput by alleviating both GPU kernel launch overhead and memory bandwidth bottlenecks. In contrast, using float32 on a single NVIDIA A6000 GPU leads to near-100% GPU utilization yet results in significant queuing of computational tasks: many operations must wait for earlier ones to finish, considerably slowing down the overall editing process. Importantly, **all methods** in our experiments are evaluated under the same bfloat16 setting to ensure a fair and consistent comparison.
>
> **Weakness 2**
>
> We hereby clarify our novelty more explicitly. We propose **FastEdit**, which applies regularization **only** to the low-rank primary semantic subspace—where most pre-edit knowledge is concentrated—while leaving the remaining directions in the key space unregularized and freely editable. This design steers new edits into the unregularized subspace, thereby better preserving pre-trained knowledge in the core semantic subspace (Table 6, Figures 1, 12 and 13). Simultaneously, it enables efficient computation via the Sherman–Morrison–Woodbury identity.
>
> In the initial submission, we primarily emphasized FastEdit’s efficiency. In this revision, we also investigate whether low-rank regularization can also improve the editing performance of full-rank baselines. Originally, we applied our method only to MEMIT; now, we extend it to EMMET and AlphaEdit, denoting the low-rank variants as **MEMIT-F**, **EMMET-F**, and **AlphaEdit-F**, respectively (Section 5.1). Their editing performance is reported in Section 5.2. Moreover, we extend our evaluation from 2,000 sequential edits (in the original paper) to **5,000–10,000 edits**. Results show that the low-rank variants consistently outperform their full-rank counterparts while maintaining high editing speed.

---

> ### Author Response · Authors · 2025-11-27
>
> **Weakness 3**
>
> To the best of our knowledge, only one recent paper O-EDIT [1] discusses orthogonality-based interference measures. However, their metric (Section 5.3, page 9) quantifies **conflict with previous edits**, i.e., whether a new edit interferes with earlier ones.
> In contrast, our safety metric $s_t$ measure **interference with the primary semantic subspace $U$**—a low-dimensional approximation of the pre-editing knowledge. Directly measuring interference with the full pre-editing key matrix $K_0 \in \mathbb{R}^{d \times (4\times10^7)}$ is infeasible due to its enormous size; instead, we use $U$ that is estimated by the LR+D factor model, as a tractable proxy that captures the dominant semantic directions. This allows us to assess whether a new update risks corrupting the core pretrained knowledge, serving as a safty metric.
>
> **Weakness 4**
>
> We acknowledge that in batch editing settings—where many edits are applied simultaneously—the advantage of FastEdit is reduced, since the matrix inversion is performed once per batch rather than per edit. We have declared this in Section 6.
> However, in the context of sequential editing, a substantial portion of the literature focuses on applying edits one after another with batch size = 1. Representative works include RECT, PRUNE, AdaEdit, SimIE [2], RLEdit [3], and others [1,4,5,6]. By contrast, a batch size as large as 100 is relatively uncommon in this line of research. To assess how well models retain knowledge under frequent updates, bs = 1 makes it  more direct to stress-test edit accumulation and long-term model stability.  Given the above, we believe our approach makes a timely contribution to this direction.
>
> **References**
> [1] O-EDIT: Orthogonal Subspace Editing for Language Model Sequential Editing. arXiv, 2024.
> [2] Towards Lifelong Model Editing via Simulating Ideal Editor. ICML, 2025.
> [3] Reinforced Lifelong Editing for Language Models. ICML, 2025.
> [4] Resolving Lexical Bias in Model Editing. ICML, 2025.
> [5] Mitigating Heterogeneous Token Overfitting in LLM Knowledge Editing. ICML, 2025.
> [6] WikiBigEdit: Understanding the Limits of Lifelong Knowledge Editing in LLMs. ICML, 2025.

---

### Official Review · Reviewer_CCmG · 2025-10-30

**Soundness:** 3
**Presentation:** 3
**Contribution:** 3
**Rating:** 4
**Confidence:** 3

**Summary:**

The paper presents FastEdit, a method to fix the speed bottleneck in model editing. State-of-the-art methods are slow due to large $O(d^3)$ matrix inversions. The authors' key insight is to assume the pre-edit knowledge representation follows a Low-Rank plus Diagonal (LR+D) structure. This assumption allows them to replace the standard regularizer with a structured one that can be inverted efficiently using the Sherman-Morrison-Woodbury (SMW) identity, dropping the computational complexity to $O(dr^2)$. They also use SVD-based compression for sequential editing and a fused covariance estimate to cut down on pre-computation time. Experiments show FastEdit is about 10x faster than baselines on LLaMA-8B and achieves comparable or better editing performance.

**Strengths:**

1. The paper tackles a critical, practical bottleneck. Current editing times are a major blocker for real-world use. A 10x speedup is a significant engineering contribution that makes real-time editing much more feasible.

2. The core technical idea is elegant. Using the expected LR+D structured regularizer to enable the SMW identity is a smart and principled way to achieve the speedup, moving beyond just brute-force computation.

3. The fused covariance estimation is another important practical win. The 24+ hour pre-computation of prior work is a huge hidden cost, and reducing it to minutes by combining a data-driven estimate with a structural prior is a big step for deployability.

4. The safety analysis in Section 5.3, with the $e_t$ and $s_t$ metrics, provides a nice geometric intuition for why some sequential editing methods fail over time, adding a good diagnostic tool to the paper.

**Weaknesses:**

1. The novelty feels a bit thin. This seems less like a new framework and more like applying a standard low-rank approximation to the covariance matrix in the MEMIT objective. Using SMW for this is a classic linear algebra trick.

2. The sequential editing comparison looks like a strawman. As described in Appendix C.2 the baselines are adapted to accumulate all past keys making their matrix inversion rank grow linearly. This guarantees they will be slow.

3. The periodic SVD compression for sequential editing is a lossy heuristic. There's no analysis of its impact on catastrophic forgetting. For instance does the model forget edit #1 after 2000 edits?

4. The results are a clear speed-accuracy trade-off. FastEdit is competitive but it doesn't win on accuracy. It underperforms PMET on Llama-3 Generality for example.

**Questions:**

1. Regarding Appendix C.2 am I understanding correctly that you made the baselines accumulate all past keys into the matrix for inversion? This seems to create an unfair comparison. Why not compare against a standard incremental SMW update for the baselines as well?

2. Did you test for catastrophic forgetting caused by your periodic SVD compression? I'm curious what the accuracy on the _first_ 100 edits is after all 2000 edits are finished.

3. How did you choose the crucial $r_0$ rank for the Llama-3 and GPT experiments? The appendix only shows a sensitivity analysis for GPT2-XL but this parameter seems central to the method's performance.

---

> ### Author Response · Authors · 2025-11-27
>
> We sincerely thank the reviewer for their positive and encouraging comments.  Below, we provide additional clarifications in response to the reviewer’s concern.
>
> **Weaknesses 1**
>
> We hereby clarify our novelty more explicitly. We propose **FastEdit**, which applies regularization **only** to the low-rank primary semantic subspace—where most pre-edit knowledge is concentrated—while leaving the remaining directions in the key space unregularized and freely editable. This design steers new edits into the unregularized subspace, thereby better preserving pre-trained knowledge in the core semantic subspace (Table 6, Figures 1, 12 and 13). Simultaneously, it enables efficient computation via the Sherman–Morrison–Woodbury identity.
>
> In the initial submission, we primarily emphasized FastEdit’s efficiency. In this revision, we also investigate whether low-rank regularization can also improve the editing performance of full-rank baselines. Originally, we applied our method only to MEMIT; now, we extend it to EMMET and AlphaEdit, denoting the low-rank variants as **MEMIT-F**, **EMMET-F**, and **AlphaEdit-F**, respectively (Section 5.1). Their editing performance is reported in Section 5.2. Moreover, we extend our evaluation from 2,000 sequential edits (in the original paper) to **5,000–10,000 edits**. Results show that the low-rank variants consistently outperform their full-rank counterparts while maintaining high editing speed.
>
> **Weaknesses 2 and Question 1**
>
> There appears to be a misunderstanding regarding our baseline implementation in Appendix A.2. We adapt the baselines (e.g., MEMIT, EMMET) to accumulate all past edit keys—i.e., using $K_0 K_0^\top + K_{[1:t]} K_{[1:t]}^\top$ for the $t$-th edit—because this is essential for stable sequential editing. Without accumulation (i.e., using only $K_0 K_0^\top + K_t K_t^\top$ for the $t$-th edit), the baselines suffer catastrophic editing failure: their edit accuracy drops to near zero after just a few hundred edits (see Figure 4 in [1], Figure 3(a) in [2], and Table 1 in [3]). Early batch-editing methods like ROME, MEMIT, and EMMET were designed for one-shot updates where all edits are known in advance; they do not generalize to the sequential editing setting.
>
> Also, accumulating past keys does not make the baselines slower. Both formulations—$(K_0 K_0^\top + K_t K_t^\top)$ and $(K_0 K_0^\top + K_{[1:t]} K_{[1:t]}^\top)$—result in $O(d^3)$ inversion cost per edit (recall that $K_0 \in \mathbb{R}^{d \times (4\times10^7)}$ is fixed and dominates the dimensionality). In contrast, FastEdit operates with $O(dr_0r_{\max})$ inversion cost via SMW, where $r_0$ and $r_{max}$ is the number of dominant directions in $K_0$ and $K_{[1:t]}$, repspectively.
>
>
> **Weaknesses 3 and Question 2**
>
> We evaluate the fine-grained editing performance of MEMIT and its low-rank regularized variant, MEMIT-F (the maximum rank \( r_{\text{max}} \) is set to 3000), over 5,000 sequential edits on the ZsRE dataset using LLaMA3. After applying all 5,000 edits, we assess the edit efficacy (**Eff.**) of the edited models on five distinct subsets: edits 1–1,000, 1,001–2,000, 2,001–3,000, 3,001–4,000, and 4,001–5,000. The results are summarized in the table below. For reference, the overall edit efficacy across all 5,000 edits is 86.8% for MEMIT and 98.8% for MEMIT-F. We also include MEMIT-F*, a variant of MEMIT-F that applies low-rank regularization without constraining the maximum rank \( r_{\text{max}} \); as shown in Table 2, it achieves an efficacy of 99.1%.
> | method       | [1:1000]  | [1001:2000] | [2001:3000] | [3001:4000] | [4001:5000] |
> |------------|-----------|-------------|-------------|-------------|-------------|
> | MEMIT     | 0.7522    | 0.8035      | 0.8717      | 0.9286      | 0.9834      |
> | MEMIT-F    | 0.9672    | 0.9837      | 0.9964      | 0.9946      | 0.9973      |
> | MEMIT-F*    | 0.9788    | 0.9912      | 0.9960      | 0.9958      | 0.9955      |
>
> We observe that all methods exhibit some degree of forgetting, with MEMIT showing a more pronounced decline in performance on earlier edits. However, (just our opinion) we think this forgetting of prior edits is not the primary concern of current model editing research. A more severe issue is **over-editing–induced model collapse**: as demonstrated in Figure 2, applying a large number of sequential edits (e.g., 10000) can corrupt the model (see the **SST** metric in Table 2).
>
> [1] WikiBigEdit: Understanding the Limits of Lifelong Knowledge Editing in LLMs. ICML, 2025.
>
> [2] Model Editing at Scale Leads to Gradual and Catastrophic Forgetting. ACL, 2024.
>
> [3] AlphaEdit: Null-Space Constrained Knowledge Editing for Language Models. ICLR, 2025

---

> ### Author Response · Authors · 2025-11-27
>
> **Weaknesses 4**
>
> We observe little  speed–accuracy trade-off, as the FFN key space is inherently low-rank—regardless of whether the keys are from pre-edit or editing inputs. This is further supported by the hyperparameter study on the maximum allowable rank $r_{\text{max}}$ for edited keys (Section 5.3) and $r_0$ (Appendix B.4). The computational complexity of our editing procedure is $\mathcal{O}(dr_0r_{\text{max}})$. By setting both $r_{\text{max}}$ and $r_0$ to approximately 2,000$\sim$5,000 for GPT-J (i.e., roughly $\frac{1}{4}d$), our method achieves faster runtime compared to the full-rank approach, which has a complexity of $\mathcal{O}(d^3)$. For reference,  $d = 16,384$ in GPT-J.
>
> **Question 3**
>
> We provide a detailed analysis of the parameter $r_0$ in Appendix B.4, where we examine its impact across all low-rank variants—MEMIT-F, EMMET-F, and AlphaEdit-F—on all models (LLaMA3, GPT-J, GPT2-XL) and datasets (CounterFact and ZsRE). Our results show that with an appropriately chosen $r_0$,  these low-rank variants can outperform their full-rank counterparts. Moreover, on the CounterFact dataset using LLaMA3, both EMMET and AlphaEdit fail to sustain 2,000 sequential edits, whereas their low-rank variants succeed when equipped with a suitable $r_0$.

---

### Official Review · Reviewer_BLSK · 2025-10-31

**Soundness:** 3
**Presentation:** 2
**Contribution:** 3
**Rating:** 6
**Confidence:** 4

**Summary:**

This paper presents FastEdit, a fast and structured model editing framework for large language models. FastEdit exploits the empirical low-rank structure of FFN key representations within Transformers, introducing a regularization scheme that enables efficient closed-form updates via the Sherman-Morrison-Woodbury (SMW) identity. The method achieves a dramatic reduction in both computational and memory costs without sacrificing edit precision or generalization. Experimental results across several benchmarks and models (GPT2-XL, GPT-J, Llama-3) show that FastEdit supports rapid, scalable sequential editing while maintaining model robustness and editing safety.

**Strengths:**

- A central strength is FastEdit's dramatic acceleration of large-scale editing—Figure 1 highlights orders-of-magnitude time reduction for performing 2,000 sequential edits, reducing editing latency from many hours to under two for state-of-the-art LLMs. This is a genuine practical advance addressing a severe real-world bottleneck in model editing.
- Table 1 and Table 2 deliver comprehensive comparisons on CounterFact and ZsRE. FastEdit matches or outperforms strong baselines (e.g., MEMIT, AlphaEdit, PRUNE) on editing efficacy, specificity, and generalization—in many cases closing the efficiency gap without loss of edit success. The wider applicability is supported by experiments on three model architectures.
- The adaptation of all baselines for sequential editing is clearly described, supporting meaningful comparison and reproducibility.

**Weaknesses:**

- The SMW-based approach relies on good estimation of the LR+D structure from a (now-small) sample of pre-edit keys. While Section 4 and Appendix A provide justification for the low-rank model, the main paper does not thoroughly analyze what happens if the pre-edit data is excessively sparse/noisy, or if the singular value spectrum is not strongly decaying—the potential for bias or regularizer miscalibration is not deeply probed.
- Some of the notation describing periodic spectral compression and SVD-based rank truncation in Algorithm 1 (Appendix B) are terse and could be more explicitly connected to the mathematical formulation in the main sections. For instance, the reuse of symbols for compressed key matrices may confuse practitioners unfamiliar with the area.

**Questions:**

Same as Weaknesses.

---

> ### Author Response · Authors · 2025-11-27
>
> We sincerely thank the reviewer for their positive and encouraging comments.  Below, we provide additional clarifications in response to the reviewer’s concern.
>
> **Weaknesses 1**
>
> Thank you for raising this important point. How does the precomputation step affect the LR+D estimation? Or equivalently, how does the precomputation step affect the downstream editing performance?
>
> First, it’s worth separating two things: (1) the cost of precomputation (i.e., collecting pre-edit keys), and (2) the speed of sequential editing. Even if we use the same large-scale precomputation as prior work—i.e., sampling ~40 million keys from Wikipedia (from 100k sentences of average length 400)—our method still gets a big speedup during the sequential editing procedure thanks to the SMW-based update.
>
> So how does the precomputation step affect the downstream editing performance? Recall that, to compensate for limited or noisy pre-edit data, we fuse the empirical covariance (estimated from sampled keys) with a prior covariance derived from the FFN down-projection matrix. This introduces a fusion hyperparameter $\alpha$. The trade-off between sample size and $\alpha$—and in particular, how $\alpha$ greatly improves robustness under limited sampling—is illustrated in the performance heatmaps in Appendix B.4.
>
> Regarding the low-rank assumption itself: across all models we tested (GPT, LLaMA, Qwen), we have not observed any FFN hidden activation whose singular value spectrum is not long-tailed. This low-rank structure has been reported in many prior works and is consistently confirmed in our experiments. Figures 5, 6 and 7 show the long-tailed singular value distribution of the pre-edit keys, and Figure 9 shows that of the editing keys. In Figure 6 (GPT-J), for example, the top 1 singular value (see the x-axis) alone accounts for at least 10% of the total spectral energy—i.e., $s_1 / (s_1 + \dots + s_{16384}) \geq 0.1$—despite there being 16,384 dimensions.
>
> **Weaknesses 2**
>
> We thank the reviewer for this helpful suggestion. In response, we have completely revised the algorithm (now in Section G) to ensure full consistency with the mathematical notation in Sections 4 and 5. Specifically:
> - We now use distinct symbols: $K_{buff}$ for incoming edit keys, $K_{comp}$ for the compressed history, and $K_{all}$ for their concatenation used in the SMW update.
> - Each step explicitly references the corresponding equations (e.g., Eq. 8 for the SMW inversion).
> - We also added brief inline comments in the algorithm to clarify the role of each variable.
>
> These changes make the connection between the algorithmic procedure and the theoretical formulation much more transparent, especially for readers less familiar with this area.

---

### Official Review · Reviewer_ES6g · 2025-11-01

**Soundness:** 2
**Presentation:** 3
**Contribution:** 2
**Rating:** 4
**Confidence:** 4

**Summary:**

The paper targets the practical efficiency bottlenecks of model editing in LLMs. It proposes FastEdit, which exploits an inherent low-rank-plus-diagonal (LR+D) structure in the edit update to accelerate computation. Specifically, the regularization term is rewritten using a structural prior $U U^\top + D$, which enables efficient inverses via the Sherman–Morrison–Woodbury identity. For sequential edits, the method maintains a low computational cost by periodically compressing the accumulated keys to keep the low rank. Experiments indicate that it can achieve faster editing while maintaining editing quality comparable to that of prior editors.

**Strengths:**

1. Although low-rank modeling and SMW are established, the paper integrates them in the model editing setting to deliver tangible speedups without changing the editing quality.
2. Leveraging the FFN down-projection SVD as a prior to estimate the key covariance is a clever way to reduce the number of samples and preprocessing time.

**Weaknesses:**

1. In many editors, the dominant per-edit cost stems from optimizing the value vectors $V$, and the inverse is often a smaller fraction. FastEdit primarily accelerates the inverse step.

2. As the number of edits grows, the rank of accumulated keys may increase. While periodic compression is proposed, the paper does not yet demonstrate that performance remains reliable at larger scales.

3. The method introduces several hyperparameters. The combined sensitivity and robustness of outcomes to these choices is insufficiently characterized.

**Questions:**

1. Please evaluate substantially more edits to test whether periodic compression continues to preserve efficacy.

2. How exactly is ''editing time'' measured—end-to-end (including data loading and forward), per-edit from the first optimization step, or only the inverse step? More details would help highlight the method's contribution.

3. How does FastEdit’s editing time compare to the parameter-preserving methods like GRACE? For sequential editing, SimIE should also be included, as it has demonstrated strong performance.

---

> ### Author Response · Authors · 2025-11-27
>
> We sincerely thank the reviewer for their thoughtful and constructive feedback. Below, we address each concern in detail.
>
> **Weakness 1**
>
> The reported editing time spans from the first to the last edit, excluding data/model loading and post-editing evaluation. The per-edit cost comprises three steps: (1) computing the key $k$, (2) computing the value $v$, and (3) computing the update matrix. Step (1) is negligible in cost. Step (2) is performed only at the final editing layer, whereas step (3) must be executed for **every** edited layer (e.g., 6 layers for GPT-J, following standard practice). Consequently, the dominant bottleneck lies in the matrix inversion required in step (3), which scales as $O(d^3)$ with respect to the hidden dimension $d$. While this remains tractable for GPT2-XL ($d = 6400$), it becomes prohibitively expensive for larger models such as GPT-J ($d = 16384$) and LLaMA3 ($d = 14336$).
>
> **Weakness 2 and Question 1**
> The accumulated editing keys $K_{[1:t]} \in \mathbb{R}^{d \times t}$ are inherently low-rank because both pre-editing and editing keys reside in the same FFN hidden space (Figure 9). This intrinsic low-rank structure allows us to cap the effective rank at $r_{\text{max}}$ without compromising performance—mirroring the observation that even the $K_0$ (of size $d \times (4\times10^7)$) is effectively low-rank despite its enormous scale. As shown in Figure 4, an $r_{\text{max}}$ of 3000 suffices for both 5,000 and 10,000 sequential edits.
>
> We evaluate FastEdit under **2,000 to 10,000 sequential edits** (Tables 1–2, Figure 2). At 5,000 edits, for example, FastEdit completes the entire sequence in approximately 4–5 hours (Figure 3) and consistently outperforms the full-rank baselines. We attribute this to our low-rank regularization strategy: it regularizes only the low-rank primary semantic subspace—where most pre-edit knowledge is concentrated—while leaving the remaining directions in the key space unregularized and freely editable. This design steers new edits into the unregularized subspace, thereby better preserving pre-trained knowledge in the core semantic subspace (Table 6, Figures 1, 12 and 13). Simultaneously, it enables fast computation via the Sherman–Morrison–Woodbury identity.
>
> **Weakness 3**
>
> All hyperparameters (including $r_0$, $r_{\max}$, and the prior fusion coefficient $\alpha$) and their sensitivity are detailed in Appendices B.3 and B.4. Our ablations show FastEdit is robust across a wide range of settings.
>
> **Question 2**
> Editing time is measured **end-to-end**, from the start of the first edit to the completion of the last edit, excluding any post-editing evaluation (as clarified in Appendix B.3). The reported speed may appear surprisingly fast—e.g., completing 5,000 sequential edits on LLaMA3 in under 4 hours—but please note that we use the **bfloat16** precision. For 6–8B-scale models like LLaMA3, bfloat16 substantially boosts computational throughput by alleviating both GPU kernel launch overhead and memory bandwidth bottlenecks.
> In contrast, using float32 on a single NVIDIA A6000 GPU leads to near-100% GPU utilization yet results in significant queuing of computational tasks: many operations must wait for earlier ones to finish, considerably slowing down the overall editing process. Importantly, **all methods** in our experiments are evaluated under the same bfloat16 setting to ensure a fair and consistent comparison.
>
> **Question 3**
> We compare FastEdit against GRACE, SERAC, MELO, WISE, and SimIE in Appendix C.4. GRACE is fast during editing but incurs inference latency due to its dynamic key lookup and value replacement mechanism. SimIE protects past edits using a general mapping function—a strategy fundamentally distinct from ours (see Appendix A.2)—but this introduces additional per-edit computational overhead. In contrast, FastEdit achieves the best overall trade-off: it attains the highest editing accuracy while maintaining the lowest end-to-end editing time (including the post-editing evaluation) among all evaluated methods.
> Moreover, FastEdit is not mutually exclusive with SimIE, which could also potentially benefit from our low-rank regularization to enhance editing performance and accelerate computation—suggesting potential for future integration.

---

### Author Response · Authors · 2025-11-27
**Paper Revision**

We thank the reviewers for their helpful comments. In the original submission, we primarily emphasized the **computational efficiency** benefited from low-rank regularization. In this revision, we further highlight its **enhanced editing performance**: by regularizing **only** the low-rank primary semantic subspace and leaving the remaining directions in the key space unregularized and freely editable, our design channels edits into the unregularized subspace, thereby better preserving pre-trained knowledge in the primary semantic subspace.
To reflect this, we have:
- Expanded Section 5 to analyze how low-rank variants (MEMIT-F, EMMET-F, AlphaEdit-F) better preserve pre-trained knowledge and outperform full-rank baselines.
- Extended sequential editing experiments from 2,000 to 5,000–10,000 edits, showing the scaling behavior of full-rank and low-rank editing methods.
- Added supporting results in the appendix (e.g., implementation details, hyperparameter studies).
- Added a downstream GLUE task (Stanford Sentiment Treebank) to evaluate the general capability of the edited model, serving as an additional test of the edit locality.

Other changes include a more detailed analysis of the edit complexity using SMW (Section 4.1) and a discussion of limitations and future work (Section 6).

All changes are highlighted in blue text for easy navigation. We appreciate the reviewers' time and effort in evaluating our revised manuscript.

---

### Author Response · Authors · 2025-12-03
**Summay of Response (1/2)**

We appreciate the reviewers’ thoughtful feedback. Their concerns are largely centered on four key aspects: (1) novelty, (2) metrics for editing efficiency, (3) scalability with increasing numbers of edits, and (4) hyperparameter analysis. Below, we address each point concisely and substantively.  For more detailed responses, please see our individual replies to the reviewers.

---

### 1. **Novelty**

One reviewer questioned whether our contribution is merely an engineering speed-up rather than a scientific advance. We respectfully disagree. While our initial submission emphasized *editing efficiency*, the revised version further highlights how **low-rank regularization substantially improves editing performance**—in terms of both accuracy and stability. Our method applies regularization **only** on the primary semantic subspace U to protect the core pre-trained knowledge, while leaving the remaining directions unregularized and freely editable. This design not only accelerates editing but—more importantly—**significantly outperforms full-rank regularization baselines in editing accuracy and stability**.

We validate this through extensive experiments in the revised manuscript (Section 5.1):
- On LLaMA3-8B with 2,000 sequential edits on CounterFact, full-rank methods (AlphaEdit, EMMET) achieve **0% editing success (Eff\*)**, whereas their low-rank variants (**AlphaEdit-F**, **EMMET-F**) exceed **90% accuracy**.
- Downstream evaluation on sentiment analysis shows that low-rank regularization preserves general task performance, while full-rank methods suffer severe degradation—indicating catastrophic interference with retained knowledge.
- Our proposed safety metric $||\Delta U||$, which quantifies interference with the core subspace U, is significantly lower for low-rank methods, providing empirical evidence of better knowledge preservation.

Thus, our work proposes a **principled architectural approach to model editing**—not merely an optimization for faster computation.


---

### 2. **Editing Efficiency**

Editing efficiency is measured as total wall-clock time from the first to the last edit. On LLaMA3:
- Our method completes **2K / 5K / 10K** sequential edits in **1.5 / 3.9 / 8.1 hours**, respectively—**4–5× faster** than full-rank baselines (e.g., MEMIT, more than 32 hours for 10K edits) and **>10× faster** than post-regularized methods like PRUNE.

The reviewer questioned whether such speedups are plausible, pointing out that value vector computation dominates cost. However, they overlook a key detail:
- The **value vector** is computed **only once** (for the final layer), while the **update matrix** must be computed **for every editing layer** (e.g., layers {4–8} in LLaMA3).
- For high-dimensional keys ($ d = 14,336 $ in LLaMA3), full-rank baselines involve $ O(d^3) $ matrix inversions per layer. In contrast, our low-rank approach reduces complexity to $ O(d r_0 r_{\text{max}}) $, where $ r_0, r_{\text{max}} \approx \frac{1}{4}d \sim \frac{1}{3}d $.
- Empirically: computing the update across 5 editing layers takes **~8 seconds** with full-rank methods but **<0.5 seconds** with ours—while the final value vector costs only **~2.5 seconds**.

---

### 3. **Scalability**

A concern was raised that accumulating edits would increase the rank of edited keys, eroding our low-rank advantage. We counter that **both pre-trained and edited key matrices are inherently low-rank** (they are from the same key space), as evidenced by their long-tailed singular value spectra.

Consequently:
- The sampled pretrained key matrix $ K_0 \in \mathbb{R}^{d \times 4 \times 10^7} $ can be low-ranked by $ U \in \mathbb{R}^{d \times r_0} $.
- The edited key matrix $ K_1 \in \mathbb{R}^{d \times 10^4} $ (10,000 edits) can be low-ranked by $ U' \in \mathbb{R}^{d \times r_{\text{max}}} $.

This structural property ensures scalability:
- At 10K edits, our method finishes in **8 hours** vs. **>32 hours** for baselines, while achieving higher editing accuracy.

---

### 4. **Hyperparameter Investigation**

We thoroughly analyze key hyperparameters:
- **Rank parameters** $ r_0 $ and $ r_{\text{max}} $: Setting $r_0$  to $ \frac{1}{5}d\sim \frac{1}{4}d$ matches full-rank performance; $ \frac{1}{4}d \sim d $ **outperforms** full-rank methods; using $ r_0 = d $ recovers the full-rank baseline. Setting $ r_{\text{max}} = \frac{1}{5}d $ is sufficient, and increasing it further yields no noticeable gains.
- **Precomputation**: Unlike prior work requiring >24 hours, our fast precomputation takes only **minutes**. The editing performance with respect to the sampling number and the fusion coefficient $\alpha$ are invesigated.

---

### Author Response · Authors · 2025-12-03
**Summay of Response (2/2)**

We also note that Reviewer *jfow* assigned our paper a score of **6** (accept), though their review was mistakenly posted on another paper (https://openreview.net/forum?id=iLWxh1RSEd). We hope this positive assessment can contribute to a more balanced overall evaluation.

For reproducibility: our low-rank regularization strategy is simple to implement and reproduce—it replaces $K_0 K_0^\top$ with the low-rank plus diagonal structure $U U^\top + D$. Code is provided in the revised manuscript. Moreover, we open-source our implementations of the baselines, adapted to support sequential editing by incorporating a mechanism that protects previously applied edits (see Appendix A.2, inspired by AlphaEdit). This adaptation significantly boosts their sequential editing performance (https://anonymous.4open.science/r/me-1DF4). We consider this release important: in reviewing recent literature, we observed that some newly published works implicitly adopt the technique described in Appendix A.2—yet attribute the significant performance gains to their own proposed methods, when in fact the improvements largely stem from the technique in Appendix A.2. By publicly releasing our code and providing a clear, documented implementation in Appendix A.2, we aim to promote **fair and transparent** benchmarking in the sequential editing setting.

In summary, our work presents a **principled methodological advance**: a low-rank regularization strategy that simultaneously enhances editing *accuracy*, *efficiency*, and *scalability* over full-rank baselines. These gains are demonstrated across multiple models, datasets, and evaluation protocols. We believe our revised paper presents a valuable contribution and merits publication.

---

### Note · Authors · 2026-01-06

I have read and agree with the venue's withdrawal policy on behalf of myself and my co-authors.